# DEXTER: Diffusion-Guided EXplanations with TExtual Reasoning for Vision Models

**Simone Carnemolla**[1]* **Matteo Pennisi**[1]* **Sarinda Samarasinghe**[2]
**Giovanni Bellitto**[1] **Simone Palazzo**[1] **Daniela Giordano**[1]
**Mubarak Shah**[2] **Concetto Spampinato**[1]

[1]University of Catania     [2]University of Central Florida
simone.carnemolla@phd.unict.it     matteo.pennisi@unict.it

## Abstract

Understanding and explaining the behavior of machine learning models is essential for building transparent and trustworthy AI systems. We introduce DEXTER, a data-free framework that employs diffusion models and large language models to generate global, textual explanations of visual classifiers. DEXTER operates by optimizing text prompts to synthesize class-conditional images that strongly activate a target classifier. These synthetic samples are then used to elicit detailed natural language reports that describe class-specific decision patterns and biases. Unlike prior work, DEXTER enables natural language explanation about a classifier's decision process without access to training data or ground-truth labels. We demonstrate DEXTER's flexibility across three tasks—activation maximization, slice discovery and debiasing, and bias explanation—each illustrating its ability to uncover the internal mechanisms of visual classifiers. Quantitative and qualitative evaluations, including a user study, show that DEXTER produces accurate, interpretable outputs. Experiments on ImageNet, Waterbirds, CelebA, and FairFaces confirm that DEXTER outperforms existing approaches in global model explanation and class-level bias reporting. Code is available at https://github.com/perceivelab/dexter.

## 1 Introduction

*How can we systematically uncover and explain a deep visual classifier's decision-making process in a way that is both comprehensive and human-interpretable?*
This question is crucial as AI systems are increasingly deployed in high-stakes applications, where interpretability and trust are as important as accuracy. However, the lack of transparency in model reasoning, often exacerbated by the reliance on spurious correlations—irrelevant features that disproportionately influence predictions— undermines confidence in these systems, making decisions difficult to justify. For instance, ImageNet-trained classifiers have been shown to favor background textures or lighting conditions over intrinsic object properties [40, 9]. Addressing these issues requires explainability techniques that extend beyond local attribution, offering a global perspective on a model's reasoning patterns, biases, and learned representations.

Existing methods for model explainability, such as GradCAM [38] and Integrated Gradients [42] provide *local explanations* and require data. These methods focus on analyzing individual predictions by attributing importance to specific pixels or regions in an image. While these methods are extremely useful for highlighting areas of interest, they do not offer a *global understanding* of model behavior. In

---

*Equal contribution.

39th Conference on Neural Information Processing Systems (NeurIPS 2025).

contrast, activation maximization techniques [26–28] have been instrumental in globally visualizing the features learned by neural network, but the generated images are often abstract and challenging to interpret. While existing methods offer useful insights into model behavior and feature representations, they often fall short of capturing the *global* reasoning patterns and biases behind predictions. In particular, visual explanations can be hard to interpret and may lack the ability to convey high-level reasoning or reveal subtle spurious correlations—issues that are often better addressed through complementary textual explanations [18, 6, 16, 3].

Text-based explanations offer an accessible complement to visual methods [11], but they often lack global perspective or clarity. Recent work [16, 2, 10], including Natural Language Explanation (NLE) methods [29, 36, 15], combines visual and textual cues to reveal model biases, but typically relies on labeled data, annotations, or pretrained vision-language mappings [2].

We propose **DEXTER**, a framework that generates human-interpretable textual explanations, uncovering the global reasoning patterns and biases of visual classifiers. DEXTER operates in a **fully data-free setting** by leveraging the generative capabilities of diffusion models and the reasoning power of large language models (LLMs). At its core, DEXTER optimizes soft prompts, which are mapped to discrete hard prompts, in order to guide a diffusion-based image generation process. This ensures that the generated images are aligned with the outputs of a target classifier, capturing the features and concepts prioritized by the model. The generated images are then analyzed by an LLM, which reasons across them to provide textual explanations of the classifier's decision-making process. By combining image generation and textual reasoning, DEXTER not only overcomes the interpretability challenges of visual explanations but also provides a global understanding of model behavior, including identifying spurious correlations and biases, without requiring access to any training data or ground truth labels.

Thus, the core main contributions of DEXTER are:

- **Global explanations:** A high-level understanding of model decisions, identifying key features, biases, and patterns beyond local attribution.

- **Data-free approach:** DEXTER requires only the trained model for explanations, unlike existing methods, that employ training or ground-truth data.

- **Bias identification and explanation:** Uncovering and describing spurious correlations to support model debiasing.

- **Natural language reasoning:** LLM-generated textual explanations that enhance interpretability over purely visual methods.

We evaluate DEXTER across three tasks: (1) activation maximization to reveal model-prioritized features, (2) slice discovery to detect underperforming subpopulations, and (3) bias explanation to identify spurious correlations. Experiments use SalientImageNet [40], Waterbirds [35], CelebA [21], and FairFaces [14]—datasets widely used in fairness and interpretability research. Across tasks, DEXTER demonstrates strong performance, offering meaningful insights into model behavior.

## 2 Related work

**Activation Maximization (AM)** interprets neural networks by generating inputs that maximize neuron activations [7]. Early methods often produced unrealistic, uninterpretable images [39, 25], later improved through regularization [23] and generative priors [26]. DiffExplainer [30] introduced diffusion-based image generation guided by soft prompts, which improve realism but lack semantic transparency. This work points toward token-wise optimization, but this direction remains underexplored. In contrast, DEXTER builds on this line by replacing soft prompts with discrete (*hard*) tokens, which are interpretable and enable both visual and textual global explanations. Although once central to model interpretability, AM has seen limited progress in recent years, partly because the resulting images—while optimized for neuron activation—are often difficult to interpret semantically, making it challenging to draw clear, causal insights about model behavior. The field has largely shifted toward attribution methods—e.g., GradCAM [38], Integrated Gradients [42], which are computationally efficient and provide intuitive saliency maps, though they are inherently local and data-dependent. Other efforts in feature visualization [49] offer insights into internal representations but require training data and remain focused on visual output. DEXTER revives AM by leveraging modern

diffusion models and discrete prompt optimization to produce class-level, multimodal explanations in a fully data-free setting.

**Explanation of visual classifiers.** Textual and Multimodal Explanations seek to complement visual saliency with natural language justifications. Approaches like Multimodal Explanations [29, 36], and post-hoc counterfactuals [2] align vision and language to explain model decisions. However, these methods often rely on ground-truth labels or annotated datasets and or limited to instance-level explanations. In contrast, DEXTER provides global, class-wise explanations without supervision or access to data.

**Natural Language Explanation (NLE)** methods aim to generate human-readable justifications, typically using VQA-style benchmarks [29, 36, 15]. Recent approaches integrate vision and language into unified architectures [37, 36], but remain supervised and local. DEXTER differs by producing global textual reports through unsupervised classifier probing—without labels, data, or task-specific fine-tuning.

**Slice Discovery** identifies dataset subpopulations where a model underperforms, offering a targeted way to reveal and explain systematic failures in classifier behavior. Traditional methods analyze embeddings, gradients, or misclassifications [1, 24, 41, 48], while recent approaches leverage CLIP's joint text-visual embeddings [8, 12, 47] or use LLMs to generate captions and extract keywords [44, 16]. Bias-to-Text (B2T)[16] extracts pseudo-bias labels from misclassified image captions, and LADDER[10] generates bias hypotheses from low-confidence predictions, clustering samples via LLM-derived pseudo-attributes. Unlike these data-dependent methods, DEXTER discovers and explains biases in a fully data-free manner, relying only on the classifier's internal behavior.

## 3   Method

DEXTER's framework, shown in Fig. 1, integrates three key components: a text pipeline for optimizing prompts, a vision pipeline for the image generation process, and a reasoning module using a vision-language model (VLM). DEXTER begins by optimizing a soft prompt to condition a BERT model [5] to fill in masked tokens in a predefined sentence. The resulting prompt guides the stable diffusion process to generate images that maximize the activation of a set of target neurons (e.g. classification heads) in a given visual classifier. The generated images are then analyzed by the VLM, which reasons across multiple images to provide coherent, human-readable textual explanations of the model's decision-making process.

### 3.1   Text pipeline

The text pipeline has the goal of optimizing a textual prompt to suitably condition the diffusion model. We pose prompt generation as a masked language modeling task, and employ a pretrained and frozen BERT model for this purpose. The structure of the textual prompt to be produced is fully customizable, and can be controlled by combining portions of fixed text with a set of mask tokens. For the sake of clarity and without loss of generality, we will assume that the textual prompt has the structure of a sequence $\mathbf{t} = [\mathbf{t}_{\text{fixed}}, m_1, m_2, \dots, m_N]$, where $\mathbf{t}_{\text{fixed}}$ is the portion of fixed text and all $m_i$ are set to BERT's [MASK] token. Let $\mathbf{t}_{\text{emb}}$ be the embedding of $\mathbf{t}$, including positional encoding. In order to alter BERT's behavior, which would naturally tend to replace $m_i$ with the most likely tokens based on its pretraining, we also prepend to the input sequence a learnable soft prompt $\mathbf{p} \in \mathbb{R}^{P \times d}$, consisting of a sequence of $P$ vectors, with $d$ being the dimensionality of BERT's embedding space [17]. The full input sequence to BERT is thus $[\mathbf{p}, \mathbf{t}_{\text{emb}}]$. We read out the logits corresponding to the masked tokens, i.e., $[\mathbf{l}_1, \dots, \mathbf{l}_N]$, where each $\mathbf{l}_i \in \mathbb{R}^V$ and $V$ is BERT's vocabulary size. Each logit vector $\mathbf{l}_i$ is mapped to a differentiable one-hot vector $\mathbf{o}_i \in \{0, 1\}^V$, $\sum o_{i,j} = 1$, through a Gumbel-softmax [22, 13] (with temperature $\tau = 1$), from which the corresponding predicted token $\hat{t}_i \in \{1, \dots, V\}$ is retrieved. The resulting text prompt can be recovered as $[\mathbf{t}_{\text{fixed}}, \hat{t}_1, \dots, \hat{t}_N]$.

At this point, a practical problem arises, in that standard implementations of diffusion models (e.g., Stable Diffusion [34]) CLIP's text encoder [31] is employed to embed textual prompts into a conditioning vector. Unfortunately, BERT's and CLIP's vocabulary overlap only partially. To address this issue, we employ a translation matrix $\mathbf{M} \in \{0, 1\}^{V \times W}$ to map each one-hot vector $\mathbf{o}_i$ to its corresponding representation in CLIP's vocabulary, of size $W$. In $\mathbf{M}$, each row contains a single 1, indicating the index of the corresponding token in the CLIP vocabulary. Thus, given an original one-hot vector $\mathbf{o}_i$ provided by BERT, we can translate it into its CLIP equivalent through $\mathbf{o}_i^{(C)} = \mathbf{o}_i \mathbf{M}$, indexing a token $\hat{t}_i^{(C)}$ in CLIP's vocabulary. For unassigned BERT tokens, the corresponding rows in

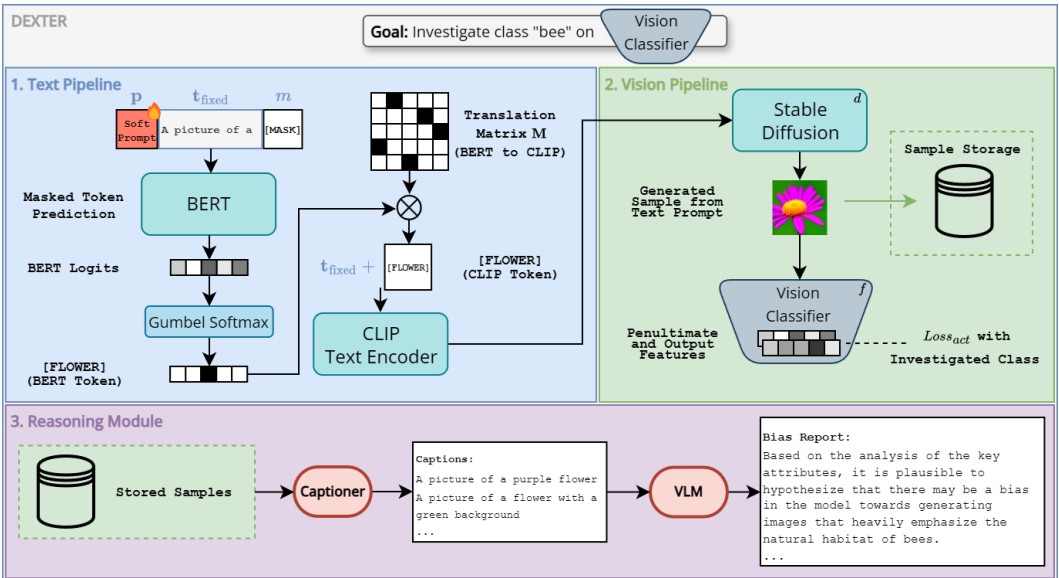

Figure 1: **DEXTER** investigates classifier biases by optimizing a learnable soft prompt to generate text prompts. These text prompts condition a diffusion model to generate images that maximize the activation of the target class in the vision classifier. Images that correctly activate the target class are stored and later captioned for Bias Reasoning. A VLM reasons using these captions to produce human-understandable textual explanations of the model's decisions and potential biases. More details and clarifications about the pipeline can be found in the Appendices A and B.

$\mathbf{M}$ are entirely zero. As a result of this design, the model learns to avoid predicting BERT tokens that do not have a valid mapping in CLIP, as they correspond to a zero indexing vector $\mathbf{o}_i^{(C)}$, which would provide a meaningless representation when multiplied by the CLIP embedding look-up table.

The textual prompt used to condition the diffusion model is thus $\left[\mathbf{t}_{\text{fixed}}, \hat{t}_1^{(C)}, \ldots, \hat{t}_N^{(C)}\right]$. Thanks to the Gumbel-Softmax activation and the linear mapping between vocabularies, the whole process is fully differentiable, allowing us to optimize the soft prompt $\mathbf{p}$, which is the only learnable parameter, through classic backpropagation.

### 3.2 Vision pipeline

The goal of the vision pipeline is to synthesize a realistic and interpretable image that maximizes the activation of a set of neurons of a visual classifier. In the following, we refer to "neurons" in a generic way, regardless of whether they encode *features* (i.e., neurons in intermediate layers) or *classes* (i.e., neurons in the output layer, after applying softmax). Given the predicted textual prompt $\left[\mathbf{t}_{\text{fixed}}, \hat{t}_1^{(C)}, \ldots, \hat{t}_N^{(C)}\right]$ (with all tokens ensured to belong to CLIP's vocabulary as explained in Sect. 3.1), we feed it to CLIP's text encoder to obtain an embedding vector $\mathbf{e}$. This vector is then used to condition a pretrained and frozen diffusion model $d$. In this work, we use Stable Diffusion, due to its widespread adoption and versatility in generating high-quality images from textual descriptions. Let $f : \mathcal{I} \to \mathbb{R}^K$ be the target frozen visual classifier, pretrained over a set of $C$ classes, providing as output the responses of a subset of $K$ selected neurons, whose activations we intend to maximize. Hence, we can obtain the selected activation vector as $\mathbf{n} = f(d(\mathbf{e}))$.

The whole vision pipeline is differentiable, enabling the definition of an optimization objective for $\mathbf{n}$, which directly affects the learnable soft prompt $\mathbf{p}$ in the text pipeline.

In order to guide the generation process to describe the behavior of the visual classifier, we introduce a neuron activation maximization loss $\mathcal{L}_{\text{act}}$ that encourages learning a textual prompt and a synthetic

image that maximizes the response of the selected $\mathbf{n}$ neurons. We define $\mathcal{L}_{\text{act}}$ as:

$$\mathcal{L}_{\text{act}} = \sum_{i=1}^{K} l_{\text{act}}(n_i),\tag{1}$$

where $n_i$ is the activation of the $i$-th element in $\mathbf{n}$, and $l_{\text{act}}$ depends on whether $n_i$ is a feature neuron or a class neuron :

$$l_{\text{act}}(n_i) = \begin{cases} -n_i, & \text{if } n_i \text{ is a feature neuron} \\ -\log n_i, & \text{if } n_i \text{ is a class neuron} \end{cases}\tag{2}$$

### 3.3 Masked pseudo-labels prediction

In our preliminary experiments, we observed that, when using the activation maximization objective only, the gradient propagated to the learnable soft prompt $\mathbf{p}$ in the text pipeline was too small, slowing down or even preventing convergence. To address this issue, we introduce, in the text pipeline, an auxiliary mask prediction task to provide a shorter backpropagation path to $\mathbf{p}$. The design of the auxiliary task gives us the opportunity to add another explainability feature to our approach: associating masked tokens with subsets of target neurons, in order to encourage the mapping between neuron activations and specific portions of the textual prompt.

To facilitate this, we initialize a set of pseudo-labels $y_1, \ldots, y_N$, one for each mask token position $m_1, \ldots, m_N$. Each pseudo-label $y_i \in \{1, \ldots, V\}$ is associated with a reference loss $L_i$, initialized to $+\infty$, and with the set of reference neurons $\mathcal{N}_i \subseteq \{1, \ldots, K\}$. At each optimization step, for each mask token $m_i$, BERT predicts the logits $\mathbf{l}_i$, from which we compute the (standard) softmax vector $\mathbf{s}_i$ and the corresponding predicted token $\hat{t}_i$, following the notation introduced in Sect. 3.1. We define the aggregated activation loss $\mathcal{L}_{\text{agg},i}$ for the set of associated reference neurons:

$$\mathcal{L}_{\text{agg},i} = \sum_{j \in \mathcal{N}_i} l_{\text{act}}(n_j).\tag{3}$$

Then, if $\mathcal{L}_{\text{agg},i}$ is smaller than the corresponding reference loss $L_i$, we update both the pseudo-label $y_i \leftarrow \hat{t}_i$ and the reference loss $L_i \leftarrow \mathcal{L}_{\text{agg},i}$. If the pseudo-label $y_i$ has been set for $m_i$ (and the corresponding reference loss $L_i$ is finite), we add a cross-entropy loss term $\mathcal{L}_{\text{mask},i} = -\log s_{i,y_i}$, with $s_{i,j}$ being the $j$-th element of the softmax vector $\mathbf{s}_i$. This approach ensures that the pseudo-labels are continually refined to better align with the activation patterns of the target neurons as training progresses, while constraining the model's parameters to remain within a region of the parameter space that corresponds to meaningful, interpretable configurations. After the first iteration, where all $y_i$ are set, the overall loss function $\mathcal{L}$ is then:

$$\mathcal{L} = \sum_{k=1}^{K} l_{\text{act}}(n_k) - \sum_{i=1}^{N} \log s_{i,y_i}.\tag{4}$$

A possible issue that may arise is the prediction of outlier tokens that could randomly decrease the activation loss, which would possibly update the pseudo-labels to spurious values. To prevent this, for each masked token position $m_i$, we maintain a history of aggregated losses for each word predicted during the optimization process, using the history mean for comparison with the reference loss $L_i$. For instance, let us assume that masked token $m_i$ has been mapped by BERT to vocabulary token $t^*$ in $T$ training iterations, building up a list of corresponding aggregated losses $\left[ \mathcal{L}_{\text{agg},i}^{(1)}, \ldots, \mathcal{L}_{\text{agg},i}^{(T)} \right]$. Instead of comparing $L_i$ with the most recent aggregated loss $\mathcal{L}_{\text{agg},i}^{(T)}$, we check whether:

$$\frac{1}{T} \sum_{j=1}^{T} \mathcal{L}_{\text{agg},i}^{(j)} < L_i,\tag{5}$$

and only in this case do we update $L_i$. This approach prioritizes the prediction of words with lower historical activation loss, preventing the selection as pseudo-target outlier words that lead to random loss fluctuations.

# 4 Performance analysis

## 4.1 Datasets

To demonstrate the versatility and effectiveness of DEXTER as a global explanation framework, we design a comprehensive evaluation protocol that reflects the core dimensions of model interpretability: feature relevance, bias identification, and semantic alignment. We evaluate DEXTER across four key tasks: **visual explanation, activation maximization, bias discovery, and bias text explanation**, using four widely adopted datasets that enable rigorous and complementary assessments of interpretability. Each dataset was selected for its alignment with a specific evaluation goal and its established use in the literature.

*SalientImageNet* [40] is used to evaluate both *visual explanations and activation maximization*. It is a curated subset of ImageNet designed to analyze model reasoning through object and context annotations. For each class, the top-5 neural features—highly activated units in the penultimate layer—are annotated as either spurious or core, based on whether they reflect incidental or meaningful correlations with the target label. This enables a fine-grained assessment of DEXTER's ability to highlight robust, semantically grounded features in both explanation and feature synthesis settings.

*Waterbirds*[35] and *CelebA*[21] serve as *benchmarks for bias discovery*. *Waterbirds* introduces spurious correlations between bird species and backgrounds (e.g., land vs. water), providing a controlled environment for evaluating slice discovery and DEXTER's ability to surface underperforming subpopulations. *CelebA* offers over 200,000 annotated face images with 40 binary attributes (e.g., gender, hairstyle, glasses), allowing us to assess how DEXTER identifies biases in classifier behavior related to demographic and semantic features.

*FairFaces* [14] is finally used for bias text explanation, focusing on the articulation of systematic spurious correlations in facial classification tasks. With over 100,000 images balanced across seven demographic groups, it enables a rigorous evaluation of how DEXTER captures and communicates bias in decision-making across diverse populations.

For each evaluation task, detailed optimization strategies, training procedure, VLM prompts and other information are provided in the appendix.

## 4.2 Results

We evaluate DEXTER's *visual explanations* through quantitative metrics and a user study, then assess its performance in *bias discovery, mitigation, and explanation*, showcasing its ability to identify spurious correlations and enhance fairness. Finally, an *ablation study* examines the impact of individual components and DEXTER's effectiveness in optimizing text prompts for *activation maximization*, comparing it to existing methods.

### 4.2.1 Visual Explanations

We evaluate DEXTER's *visual explanations* through qualitative and quantitative comparisons with **DiffExplainer**, the only prior method that shares a similar diffusion-based generation pipeline, on the SalientImageNet in order to assess the semantic relevance and clarity of the synthesized explanations. For qualitative analysis, we conducted a user study (see Appendix H), involving 100 participants on Amazon Mechanical Turk, to further assess the interpretability of DEXTER's visual explanations compared to DiffExplainer's (the two methods with the highest CLIP-IQA scores in Tab. 6). Participants compared images generated by DiffExplainer and DEXTER alongside GradCAM-based attention heatmaps from SalientImageNet, evaluating similarity across three feature categories: *perceptual features* (shape, texture, color), *conceptual features* (semantics, context), and cases where no similarity was perceived (*None*). As shown in Fig. 2, DEXTER was preferred for conceptual features, while DiffExplainer was favored for perceptual attributes. Notably, DEXTER received fewer *None* responses, indicating stronger alignment with classifier attention regions. A chi-square test ($\chi^2 = 15.36$, $p = 0.032$) confirms a significant difference, with post-hoc analysis highlighting *None* and *conceptual features* as key contributors.

 Fig. 3 presents an example of visual and textual explanations generated by DEXTER for RobustResNet50 [40], focusing on the 5 most active features in the penultimate layer for the *dog sled* class, which are all categorized as spurious in SalientImageNet. DEXTER-generated images align more effectively with the classifier's attended regions compared to those generated by DiffExplainer. Furthermore, DEXTER provides textual descriptions that clearly explain the semantics of each

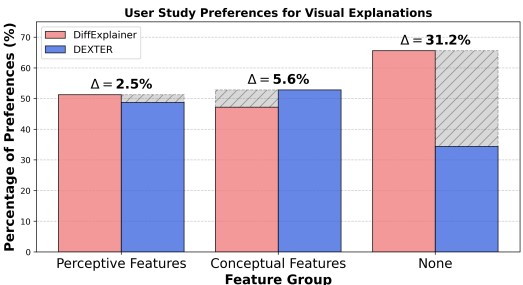

Figure 2: **User study results on SalientImageNet**: Participants evaluated the alignment between classifier attention and visual explanations, categorized into perceptual features (shape, texture, color), conceptual features (context, semantics, multiple elements), and no alignment.

neural feature, offering an interpretable account of the classifier's reasoning. This approach enhances interpretability and enables users to either identify spurious correlations, such as the ones in the *dog sled* class where none of the five most important features actually include a sled, or confirm the classifier's reliance on meaningful, task-relevant attributes.

For quantitative evaluation, we compare DEXTER to DiffExplainer using CLIP-IQA and Semantic CLIP-IQA [43] (Appendix D). DEXTER achieves higher scores (0.94 ± 0.03 / 0.96 ± 0.03) than DiffExplainer (0.89 ± 0.09 / 0.90 ± 0.05), indicating better semantic alignment and consistency. Additional comparisons with GAN-based methods [45, 23, 26] further confirm the advantage of diffusion-based explanations.

Extended details/examples on the visual explanation task are in Appendix D.

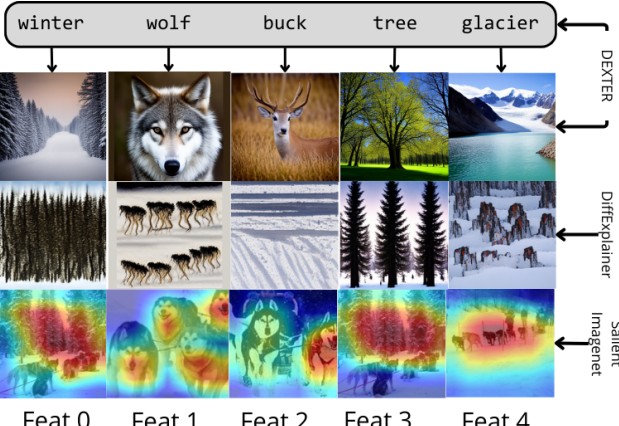

Figure 3: **DEXTER and DiffExplainer explanations** for RobustResNet50 on *"dog sled"*, showing spurious top-5 features as core visual elements.

### 4.2.2 Slice discovery and debiasing

The goal of slice discovery is to identify subgroups of data where the model exhibits worse performance compared to the rest of the dataset. We assess slice discovery and debiasing performance using CelebA and Waterbirds datasets. In the former, the task is "blonde"/"non-blonde" classification, with the "blonde" class including a low proportion of men (representing a slice). The Waterbirds dataset tackles "waterbird"/"landbird" classification, and is built so that a portion of waterbird images feature a land background, and vice versa, thus introducing a slice per class. We perform slice discovery with DEXTER by first identifying words that maximize the activation of a given class. Specifically, we leverage DEXTER's optimization process to obtain several descriptive words for each class. Following this step, as done by Kim et al. [16], we compute the CLIP similarity between the discovered words and the images in the dataset: images with high similarity are identified as belonging to a slice. We thus train a debiased classifier using Distributionally Robust Optimization (DRO) [35, 32] and evaluate the classification accuracy on the identified slices. We compare this approach with ERM [35], LfF [24], GEORGE [41], JTT [19], CNC [46], DRO [35], LADDER [10] and DRO-B2T [16]. As

shown in Tab. 1, DEXTER outperforms state-of-the-art methods on worst-slice prediction for CelebA and achieves comparable results on Waterbirds, deriving descriptive words directly from the model without relying on training data. Additional details on the slice discovery task and results are given in Appendix E.

Table 1: **Performance on slice discovery and debiasing.**

| | | | CelebA | | Waterbirds | |
|---|---|---|---|---|---|---|
| Method | Data | GT | Worst | Avg. | Worst | Avg. |
| ERM | ✓ | - | $47.7 \pm 2.1$ | 94.9 | $62.6 \pm 0.3$ | 97.3 |
| LfF | ✓ | - | 77.2 | 85.1 | 78.0 | 91.2 |
| GEORGE | ✓ | - | $54.9 \pm 1.9$ | 94.6 | $76.2 \pm 2.0$ | 95.7 |
| JTT | ✓ | - | $81.5 \pm 1.7$ | 88.1 | $83.8 \pm 1.2$ | 89.3 |
| CNC | ✓ | - | $88.8 \pm 0.9$ | 89.9 | $88.5 \pm 0.3$ | 90.9 |
| DRO | ✓ | ✓ | $90.0 \pm 1.5$ | 93.3 | $89.9 \pm 1.3$ | 91.5 |
| LADDER | ✓ | - | $89.2 \pm 0.4$ | 89.8 | $\mathbf{92.4 \pm 0.8}$ | 93.1 |
| DRO-B2T | ✓ | - | $90.4 \pm 0.9$ | 93.2 | $90.7 \pm 0.3$ | 92.1 |
| **DEXTER(Ours)** | - | - | $\mathbf{91.3 \pm 0.01}$ | 91.7 | $90.5 \pm 0.1$ | 92.0 |

### 4.2.3 Bias explanation

To evaluate DEXTER's ability to identify and explain biases in classifiers, we conduct an analysis using the FairFaces dataset [14]. Specifically, we train two binary classifiers to distinguish between two age groups: *20-29* (class 1) and *50-59* (class 2). These classifiers are trained on two variants of the FairFaces dataset: (1) a balanced dataset with equal male and female representation in both classes, (2) a dataset where males are overrepresented in class 1, and females are overrepresented in class 2. These variations allowed us to systematically assess how biases in the training data are reflected in the classifier's behavior and how well DEXTER captures these biases.

To generate bias reports for a given classifier, DEXTER produces 50 images that maximize the model's prediction for the target class. Each image is captioned using a VLM (ChatGPT-4o mini), and the list of captions is used to prompt another instance of the VLM to generate a textual bias report. This approach ensures that the generated reports reflect only the classifier's internal representations and decision-making processes (more details about VLM hallucination evaluation in appendix I). Figure 4 showcases excerpts of two generated bias reports.

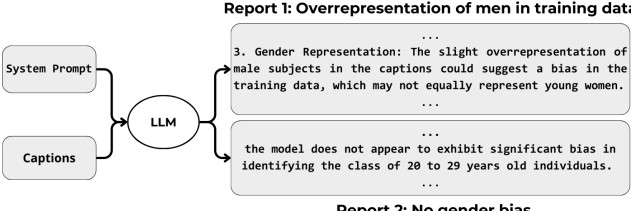

Figure 4: **DEXTER explanation reports.** *Report 1* analyzes a classifier trained on a FairFaces variant with male overrepresentation in class 1, while *Report 2* corresponds to a balanced dataset.

We evaluate DEXTER's bias reports through two complementary approaches:
**1. User Study**. In addition to the earlier study, participants evaluated and ranked DEXTER's bias reports for two classifiers. Informed of dataset biases, they assessed report accuracy, clarity, and interpretability. We report human Mean Opinion Scores ($MOS_{humans}$, 1–5 scale), and complement them with automated evaluations: $MOS_{LLM}$ from a VLM and G-eval [20] metrics, including $G\text{-eval}_{con}$ for consistency with classifier behavior. These scores jointly assess the linguistic, structural, and explanatory quality of DEXTER's outputs.
**2. Comparison with training-derived reports**. We compare DEXTER's data-free reports to those generated from training set images using the same pipeline (captioning + LLM-based reporting). Their similarity is quantified via sentence transformer similarity (STS) [33], measuring alignment with data-grounded biases.
Table 2 shows strong agreement between human ($MOS_{humans}$) and model ($MOS_{LLM}$) ratings (statistical validation in appendix F.2), with all scores across MOS and G-eval metrics exceeding 3 and

approaching 4—indicating consistent perceptions of the reports as *good* to *excellent*. High STS scores further confirm that DEXTER's reports, generated without data, align closely with training-derived insights. This demonstrates DEXTER's ability to capture model biases fluently, faithfully, and without supervision. The bias reasoning enabled by DEXTER also facilitates disparity analysis across a variety of classifiers (from transformers to CNNs) as shown in Fig. 8 of the supplementary, where we compare ViT, AlexNet, ResNet50, and RobustResNet50 on ImageNet classes.

Full methodological details and report samples are provided in Appendix F and the supplementary.

Table 2: **Evaluation of DEXTER bias reports** generated for classifiers trained on FairFaces. The columns "w Bias" and "w/o Bias" refer to models trained on datasets with and without gender bias, respectively, for each age group.

| Metric | Class 0 (20-29) | | Class 1 (50-59) | | Mean |
| --- | --- | --- | --- | --- | --- |
| | w Bias | w/o Bias | w Bias | w/o Bias | |
| STS | 0.92 | 0.85 | 0.91 | 0.91 | 0.90 |
| G-eval$_{con}$ | $4.58 \pm 1.00$ | $4.80 \pm 0.47$ | $3.66 \pm 0.84$ | $3.68 \pm 0.85$ | $4.19 \pm 0.52$ |
| MOS$_{LLM}$ | $4.29 \pm 0.63$ | $4.80 \pm 0.37$ | $4.63 \pm 0.44$ | $4.19 \pm 0.74$ | $4.48 \pm 0.25$ |
| MOS$_{humans}$ | $4.20 \pm 0.63$ | $3.89 \pm 0.64$ | $4.10 \pm 0.67$ | $3.88 \pm 0.79$ | $4.01 \pm 0.69$ |

#### 4.2.4 Ablation Study

We investigate how prompt design, auxiliary optimization, and multi-target strategies affect DEX-TER's ability to generate effective and interpretable visual explanations. Using 30 SalientImageNet classes (15 labeled with spurious and 15 with core features), we compare four prompting strategies by generating 100 images per class and measuring classifier activations: (1) class label only, (2) ChatGPT-generated descriptions, (3) captions from DiffExplainer, and (4) DEXTER's optimized prompts. As shown in Table 3, DEXTER achieves the highest mean score (75.43), effectively maximizing both spurious (63.00) and core (87.86) features. In contrast, the baseline and ChatGPT perform moderately, while DiffExplainer underperforms.

We then ablate the role of prompt length and auxiliary loss $\mathcal{L}_{mask}$ by comparing single-word vs. multi-word optimization, with and without auxiliary supervision. Single-word prompts lack expressiveness (mean 23.73), and multi-word prompts without auxiliary loss are unstable (mean 11.83). Introducing $\mathcal{L}_{mask}$ improves both cases, with the best performance achieved when combining multi-word prompts and auxiliary loss (75.43). This highlights the importance of pairing rich prompts with stable optimization to align explanations with classifier behavior. The standard deviations reported in Tables 3 and 4 may appear high due to the variation in activation strength across different classes, as some classes are consistently activated and others only partially. For instance, while some classes were activated in 100 out of 100 generations, others were only in 20 out of 100. As a result, this large variation across classes naturally leads to a high overall standard deviation. Full ablation details are provided in Appendix G.

Finally, we evaluate the faithfulness and robustness of DEXTER's explanations by testing for bias propagation and LLM-induced hallucinations. In the first test, we injected adversarial cues (e.g., using lion for tiger) into prompt initialization to assess whether upstream biases from BERT or Stable Diffusion would affect outputs. DEXTER consistently recovered class-relevant features, showing robustness to such distortions. In the second test, we extracted the most salient visual cue from each generated text report and added it to the image prompt; the resulting increase in classifier activation confirms that the cues contained in the text explanation reflect model-relevant features rather than hallucinations. Full details and statistical validation are provided in Appendix I.

## 5 Limitations

While DEXTER delivers detailed, data-free global explanations, it is computationally demanding: prompt optimization takes ~10 minutes per class, though generating 100 images and the final bias report is fast (~15 seconds without backpropagation), making it suitable for offline use. Since DEXTER relies on Stable Diffusion, there is a risk of NSFW outputs; to mitigate this, we apply its built-in safety checker to filter and discard inappropriate images during generation.

Table 3: **Comparison between text-prompting strategies** for maximizing the target class.

| Method | Spurious | Core | Mean |
|---|---|---|---|
| Baseline | $43.06 \pm 38.86$ | $86.40 \pm 23.83$ | $64.73 \pm 31.34$ |
| ChatGPT description | $41.20 \pm 40.78$ | $78.53 \pm 34.02$ | $59.87 \pm 37.40$ |
| DiffExplainer | $33.20 \pm 41.07$ | $47.66 \pm 44.80$ | $39.83 \pm 42.93$ |
| DEXTER (ours) | $\mathbf{63.00 \pm 31.20}$ | $\mathbf{87.86 \pm 15.14}$ | $\mathbf{75.43 \pm 23.17}$ |

Table 4: **Ablation results** for single-word and multi-word prompt optimization, with/without the auxiliary task $\mathcal{L}_{\text{mask}}$.

| Configuration | Spurious | Core | Mean |
|---|---|---|---|
| Single-word | $11.13 \pm 27.38$ | $36.33 \pm 38.45$ | $23.73 \pm 32.91$ |
| $\hookrightarrow + \mathcal{L}_{\text{mask}}$ | $34.00 \pm 32.72$ | $53.86 \pm 44.64$ | $43.93 \pm 38.68$ |
| Multi-word | $15.53 \pm 27.93$ | $8.13 \pm 18.74$ | $11.83 \pm 23.33$ |
| $\hookrightarrow + \mathcal{L}_{\text{mask}}$ | $\mathbf{63.00 \pm 31.20}$ | $\mathbf{87.86 \pm 15.14}$ | $\mathbf{75.43 \pm 23.17}$ |

# 6 Conclusion

We introduced DEXTER, a framework for globally explaining deep visual classifiers by combining diffusion-based activation maximization with textual reasoning. Unlike existing methods, DEXTER operates in a fully data-free setting, deriving insights solely from the classifier's internal representations. Experiments on SalientImageNet, Waterbirds, CelebA, and FairFaces demonstrate its effectiveness in uncovering spurious and core features, identifying dataset subpopulations, and generating human-readable bias explanations. Ablation studies confirmed the impact of multi-word prompts and auxiliary optimization, while user studies validated the clarity and relevance of DEXTER's textual explanations. Future work will extend its capabilities to multimodal models and on refining textual reasoning.

## Acknowledgements

Simone Carnemolla, Matteo Pennisi, Giovanni Bellitto, Simone Palazzo, Daniela Giordano, and Concetto Spampinato have been supported by the European Union – Next Generation EU, Mission 4 Component 2 Line 1.3, through the PNRR MUR project PE0000013 – FAIR "Future Artificial Intelligence Research" (CUP E63C22001940006).

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

# A Glossary

**K** - The cardinality of the set of neurons of interest, which contains both neurons in intermediate and output layers of the vision classifier

**M** - A translation matrix that maps the BERT vocabulary to the CLIP embedding space

$\mathcal{L}_{act}$ - Neuron activation maximization loss. Used to determine if the text prompts and images activate the desired neurons of the visual classifier (Eq. 1)

$\mathcal{L}_{agg}$ - Aggregated activation loss. Used also to determine if pseudolabels for $\mathcal{L}_{mask}$ (Eq. 3)

$\mathcal{L}_{mask}$ - A cross-entropy loss between the BERT [MASK] token predictions and saved pseudolabels

**p** - The learnable soft prompt in DEXTER

**t** - A sequence consisting of $t_{\text{fixed}}$ and $N$ [MASK] tokens

**t$_{\text{emb}}$** - The embedding of t, including the positional encoding

**t$_{\text{fixed}}$** - Text tokens corresponding to "A picture of a"

**V** - The BERT vocabulary size, 30522

**W** - The CLIP vocabulary size, 49408

# B  DEXTER Algorithm

---
**Algorithm 1** DEXTER
---
    **Goal:** Investigate class $c$ on Vision Classifier $f$
1: **Initialize** soft prompt **p**
2: **for** iteration = 1 to $N$ **do**
3:     Encode **p** + t$_{\text{fixed}}$ + [MASK] tokens
4:     Obtain BERT logits $l$ for masked token prediction
5:     Apply Gumbel-Softmax to $l$ to obtain the predicted token $\hat{t}$
6:     Use translation matrix **M** to convert $\hat{t}$ to the CLIP vocabulary $\hat{t}^{(C)} = \hat{t}\mathbf{M}$
7:     Compute CLIP text embedding $\mathbf{e} = [t_{\text{fixed}}, \hat{t}^{(C)}]$
8:     Generate image with diffusion model $d$ conditioned on **e**
9:     Apply cross-entropy loss $L_{act}$ with ground truth $c$ to $f(d(\mathbf{e}))$
10:     Apply auxiliary loss $L_{mask}$ on $l$ using the previous logit with the highest class activation
11:     Update **p**
12: **end for**
---

# C  Training hyperparameters

We adopt CLIP as the text encoder and Stable Diffusion v1.4 [2] as the diffusion model. To reduce inference time, we employ the Latent Consistency Model (LCM) LoRA adapter [3] using 4 inference steps. DEXTER is trained with a batch size of 1 (i.e., one image per iteration) and a learning rate of $0.1$ across all tasks. In the following sections (Appendices D, E and F), the term *optimization steps* refers to the number of iterations performed to optimize the trainable soft prompt. Each optimization step consists of: predicting the masked token, generating an image, and obtaining the visual classifier's prediction.

In the following, we detail the parameters and hyperparameter settings described in Sections 3.1 and 3.2: for the textual prompt sequence $[\mathbf{t}_{\text{fixed}}, m_1, m_2, \ldots, m_n]$, we use ``a picture of a [MASK].'' in the single-word optimization scenario, whereas for multi-word optimization, the fixed prompt is ``a picture of a [MASK] with [MASK] and [MASK] and [MASK] and [MASK] and [MASK].''
We set the sequence $P$ of soft prompts $p$ to 1 (see Sect. C.1), with each embedding having dimension $d = 768$, matching BERT's embedding space. The temperature $\tau$ for the Gumbel softmax is kept at

---

[2] Hugging Face Stable Diffusion id: `compvis/stable-diffusion-v1-4`.

[3] Hugging Face LoRA id: `latent-consistency/lcm-lora-sdv1-5`.

its default value of 1.0.

All experiments ran in half-precision on three H100 GPUs. Prompt optimization takes ∼10 minutes per class, while generating 100 images and the final bias report takes ∼15 seconds without backpropagation. DEXTER is designed for offline, global auditing, with a cost aligned to its goal of delivering comprehensive model insights.

### C.1 Number of soft prompts experiments

We set $P = 1$ to keep the setup minimal and stable. In DEXTER, the soft prompt guides BERT in selecting hard tokens via masked language modeling, rather than encoding semantics directly. Expressiveness comes from composing multi-token prompts (typically 5–6), not from prompt dimensionality. Experiments with $P > 1$ show that increasing P expands the parameter space and weakens the classifier's gradient signal, destabilizing optimization in our data-free setting, as shown in Table 5. We report mean and standard deviation for 8 randomly selected classes (4 core, 4 spurious).

Table 5: **DEXTER's performance using diffent number of** $P$.

| P | Spurious | Core | Avg |
|---|----------|------|-----|
| 1 | **81.25 ± 18.99** | **94.25 ± 4.38** | **87.75 ± 11.68** |
| 3 | 75.00 ± 29.00 | 44.50 ± 35.89 | 59.75 ± 32.44 |
| 5 | 73.50 ± 25.66 | 74.00 ± 42.75 | 73.75 ± 34.20 |
| 10 | 21.25 ± 31.17 | 67.75 ± 40.01 | 44.50 ± 35.59 |

## D  Visual Explanation Details

This appendix provides additional details on the evaluation of visual explanations generated by DEXTER, complementing the discussion presented in Section 4.2.1. Specifically, we outline the methodology behind CLIP-IQA, our proposed Semantic CLIP-IQA metric and the comparison between DEXTER and prior activation maximization works. Following, a comprehensive analysis of neural feature maximization. These insights further substantiate the findings reported in the main paper and demonstrate the robustness of DEXTER in generating interpretable visual explanations.

### D.1 Quantitative evaluation metrics: CLIP-IQA and Semantic CLIP-IQA

CLIP-IQA [43] is a metric originally introduced to evaluate image quality. In this work, we leverage CLIP-IQA to compare the quality of images generated by DEXTER against those produced by other approaches. Specifically, CLIP-IQA calculates the similarity between generated images and two fixed prompts, returning the probability that an image is closer in similarity to the first prompt rather than the second. The fixed prompts used in the original formulation are "`Good photo.`" and "`Bad photo.`"

To incorporate semantic relevance into image quality assessment, we propose **Semantic CLIP-IQA**. This metric follows the same evaluation protocol as CLIP-IQA but replaces the fixed prompts with class-specific prompts: "`Good photo of a [CLASS]`" and "`Bad photo of a [CLASS]`". This modification ensures that the evaluation captures not only general image quality but also the semantic alignment between the generated images and the target class.

We here also report in Table 6 a quantitative comparison of visual explanation quality across existing activation maximization methods. The comparison highlights the significant performance gap between older GAN-based or optimization-based approaches and diffusion-based methods. In particular, DEXTER demonstrates superior alignment and consistency, validating its ability to generate more meaningful and semantically grounded visual explanations.

### D.2 Neural Feature Maximization

*Neural feature maximization* refers our strategy to generate images that emphasize the neural features, as defined in SalientImageNet (i.e., the features of the penultimate layer of the model), learned by a neural network for a given class. By optimizing the input image to maximize the activation of

Table 6: **Quantitative comparison between DEXTER and existing activation maximization methods for visual explanations**

| Method | CLIP-IQA | Semantic CLIP-IQA |
|---|---|---|
| Yosinski et al. [45] | $0.37 \pm 0.19$ | $0.33 \pm 0.23$ |
| Mahendran and Vedaldi [23] | $0.82 \pm 0.15$ | $0.72 \pm 0.15$ |
| Nguyen et al. [26] | $0.74 \pm 0.21$ | $0.57 \pm 0.30$ |
| DiffExplainer [30] | $0.89 \pm 0.09$ | $0.90 \pm 0.05$ |
| DEXTER (ours) | $\mathbf{0.94 \pm 0.03}$ | $\mathbf{0.96 \pm 0.03}$ |

specific neural features, this method allows for an interpretable visualization of what the model has learned to recognize as distinctive for a class. This approach is useful for understanding how deep learning models make decisions and can help identify potential biases or spurious correlations in their predictions.

In our work, neural feature maximization is performed using 2,000 optimization steps with a `single word` prompting strategy (see Sect. 4.2.4). We employ RobustResNet50 as the visual classifier for this task. Table 7 reports the 30 classes selected from the SalientImageNet dataset, consisting of 15 classes containing spurious features and 15 with only core features. Specifically, we selected all 15 classes from SalientImageNet in which all top-5 features were marked as spurious. Additionally, we randomly selected another 15 classes from SalientImageNet where all top-5 features were marked as core.

Table 7: **Bias and Non-Bias Classes**

| Bias Classes | | | Non-Bias Classes | | |
|---|---|---|---|---|---|
| **Class_idx** | **Class Name** | **Bias** | **Class_idx** | **Class Name** | **Bias** |
| 706 | *patio* | ✓ | 985 | *daisy* | ✗ |
| 837 | *sunglasses* | ✓ | 291 | *lion* | ✗ |
| 602 | *horizontal bar* | ✓ | 292 | *tiger* | ✗ |
| 795 | *ski* | ✓ | 486 | *cello* | ✗ |
| 379 | *howler monkey* | ✓ | 465 | *bulletproof vest* | ✗ |
| 890 | *volleyball* | ✓ | 574 | *golf ball* | ✗ |
| 801 | *snorkel* | ✓ | 582 | *grocery store* | ✗ |
| 981 | *ballplayer* | ✓ | 635 | *magnetic compass* | ✗ |
| 746 | *puck* | ✓ | 514 | *cowboy boot* | ✗ |
| 416 | *balance beam* | ✓ | 609 | *jeep* | ✗ |
| 537 | *dogsled* | ✓ | 624 | *library* | ✗ |
| 655 | *miniskirt* | ✓ | 764 | *rifle* | ✗ |
| 810 | *space bar* | ✓ | 847 | *tank* | ✗ |
| 433 | *bathing cap* | ✓ | 879 | *umbrella* | ✗ |
| 785 | *seat belt* | ✓ | 971 | *bubble* | ✗ |

Figure 5 extends Figure 3 by reporting additional comparison between DEXTER and DiffExplainer across different classes (the full set of comparisons is provided in the supplementary). While DiffExplainer effectively maximizes shapes, colors, and textures, its outputs often lack realism, converging on abstract or pattern-like images. In contrast, DEXTER, owing to its textual anchor, more reliably yields semantically coherent representations of the features highlighted by the *SalientImageNet* heatmaps. This visually confirms the results of the user study reported in Fig. 2.

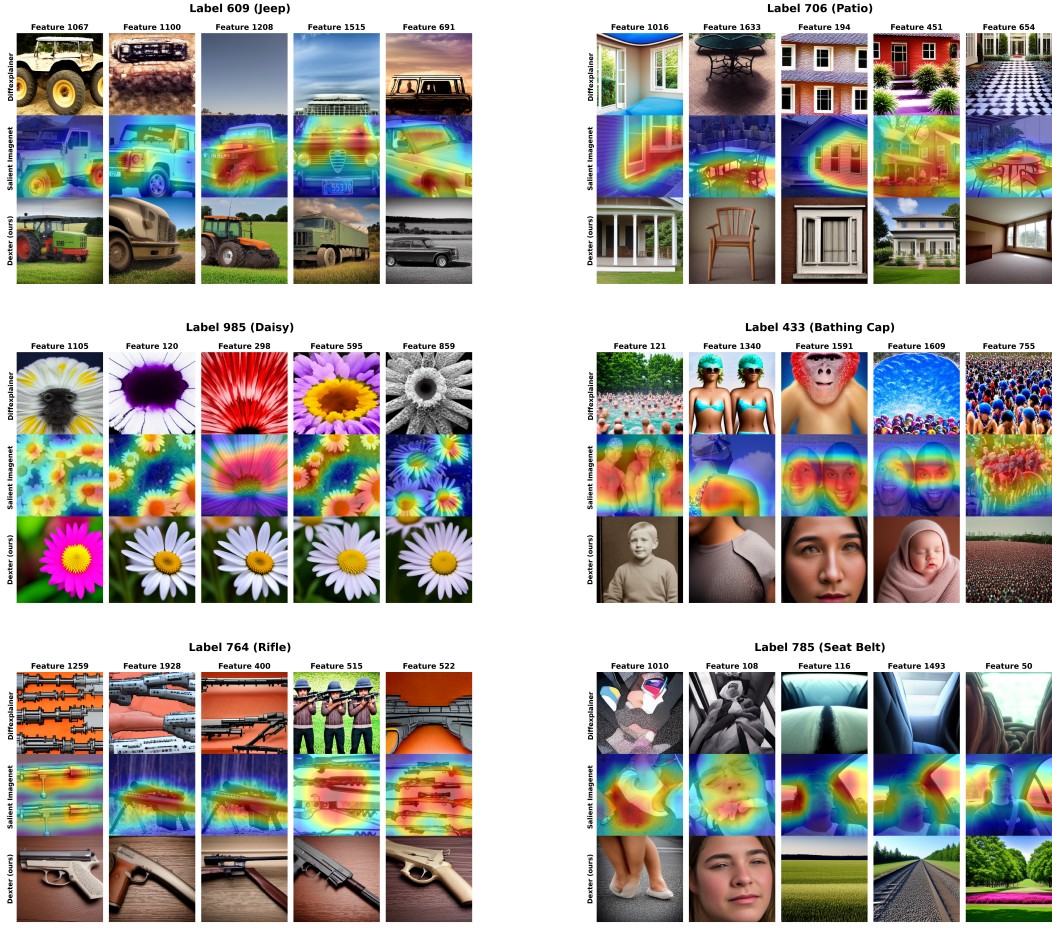

Figure 5: Comparison of *DiffExplainer* and *DEXTER* with respect to *SalientImagenet* for the neural feature maximization task, covering different classes. The left column displays classes without spurious features, whereas the right column shows classes with spurious features.

# E    Slice Discovery Details

This appendix provides additional details on slice discovery and debiasing using DEXTER, complementing the discussion in Section 4.2.2.

Figure 6 extends Figure 1 to better explain the Slice Discovery pipeline used by DEXTER. We use the Waterbirds dataset as an example, which consists of two classes: Landbirds and Waterbirds. The majority of the landbirds samples are visibly on land, and similarly the majority of the waterbirds are in/near water. While overall this is an easy task for most classifiers, we want to ensure high performance on the small amount of images where the birds are not in their natural habitats (landbirds on water, waterbirds on land).

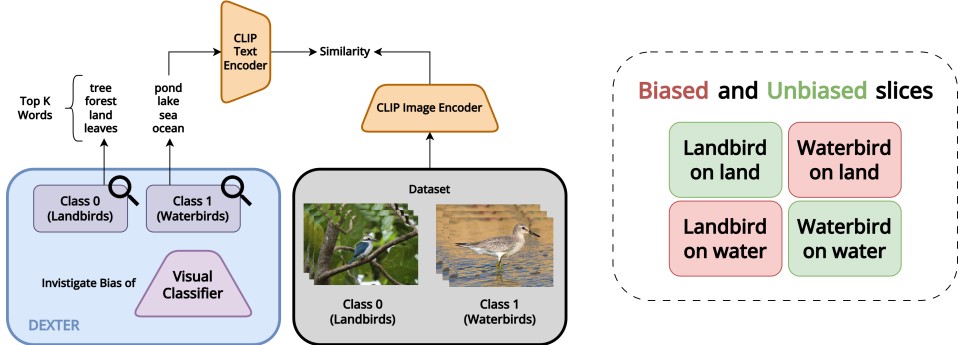

Figure 6: **Pipeline for Slice Discovery with DEXTER**. The top-k words associated with a class are encoded using the CLIP Text Encoder and compared with image encodings from the biased dataset. The appropriate slice is determined based on the similarity between text and image embeddings, along with the ground truth labels of the dataset images. The dataset is then labeled into four categories: waterbird on water, waterbird on land, landbird on land, and landbird on water (see right side).

DEXTER identifies the words explaining an input visual classifier, as presented in Sect. 3. Specifically, given a classifier $f$ and a target class $c$, it optimizes a text prompt to extract a set of top-k class words $W_C = w_{c1}, ..., w_{ck}$ (Tab. 8). Each word $w_{ci}$ is encoded using CLIP's text encoder to obtain an embedding $t_{ci}$ and the class prototype is defined as the average embedding: $\bar{t}_c = \frac{1}{k} \sum t_{ci}$ Then, following the approach in B2T [16], at inference time, each image $x_j$ of the dataset is encoded into $v_j$ using CLIP's image encoder. The similarity between the image and the class is then computed as:

$$\text{CLIP}_{\text{score}} = \text{similarity}(v_j, \bar{t}_c) \tag{6}$$

If an image has an higher similarity with the class-words that correspond to its real class it is labeled as unbiased slice (e.g. an image belonging to class landbirds with an higher similarity with landbirds class-words), while if an image has an higher similarity with its counterpart class it is labeled as biased slice (e.g. an image belonging to class landbirds has an higher similarity with waterbirds class-words). Fig. 7 presents the ROC curves obtained using the CLIP-based matching approach on the Waterbirds dataset, in comparison with other SOTA slice discovery methods [16, 8, 12]. The results illustrate how effectively the training images are partitioned into the four groups (waterbird on water, waterbird on land, landbird on land, landbird on water) by the DEXTER-derived words. Finally, we follow the debiased training scheme from [35] to train a debiased classifier on the dataset.

In all our slice discovery experiments, we set $k = 4$ both for CelebA and Waterbirds. We used 1000 DEXTER optimization steps and `single word` prompting (see Sect. 4.2.4) as an activation maximization strategy to discover the above words. Tab. 8 reports the discovered words for both datasets, where class 0 of CelebA has been left blank as in B2T [16].

## E.1    Top-k words selection details

The top-k words in Table 8 were obtained by running the DEXTER pipeline multiple times (specifically k=4 timeless). In each run, we recorded the final (single) pseudo-token selected by the masked language model at the end of the optimization. To encourage diversity across runs, we excluded previously selected words from the candidate pool before each new run. However, we also evaluated

Table 8: **Class-wise words discovered by Dexter for Waterbirds and CelebA datasets.**

| Dataset | Class 0 | Class 1 |
|---|---|---|
| Waterbirds | fence, woods, jungle, backyard | seas, sea, lake, harbor |
| CelebA | — | woman, person, head, girl |

Table 9: **Comparison between multi-run top-k word selection (ours) and single-run frequency-ranked top-k selection.**

| Method | F1-score Class 0 | F1-score Class 1 |
|---|---|---|
| B2T | **0.99** | 0.75 |
| k = 10 | 0.98 | 0.67 |
| k = 5 | 0.98 | 0.66 |
| ours | **0.99** | **0.76** |

how using sampling frequency ranking may affect slice discovery results. In particular, we computed results using word frequency–based ranking, evaluating performance with the top k = 5 and k = 10 most frequent words. We compare these results against our proposed sampling strategy and B2T for reference. As shown in Table 9, our strategy with a single word yields superior performance.

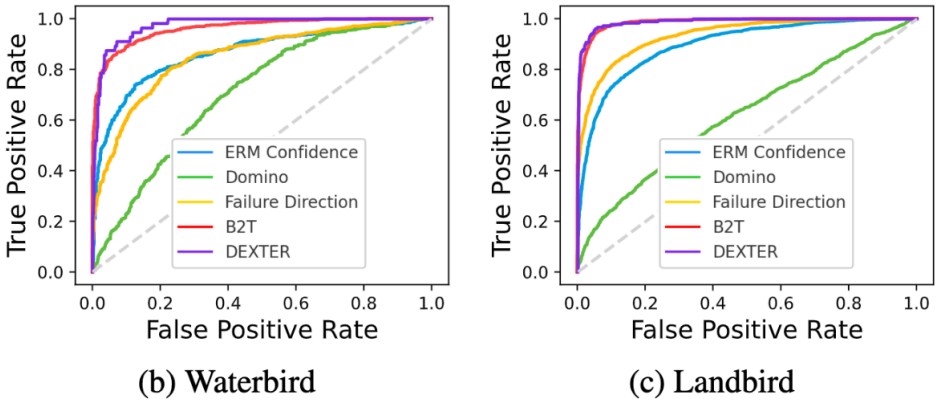

(b) Waterbird        (c) Landbird

Figure 7: ROC curves for slice discovery on the Waterbirds dataset. Each curve is obtained by comparing the slice assignments predicted by each method with the ground-truth slice labels released with the dataset.

## F  Bias Reasoning Details

Given a classifier $f$ and class $c$, we optimize a prompt $t^*$ such that a diffusion model generates images $x_1, ..., x_M$ that maximize the classifier's output for $c$. Each generated image correctly predicted by the classifier is captioned using a vision-language model $g$, producing descriptions $d_i(g(x_i))$. A language model $h$ then summarizes the captions into a single report, describing the features associated with class $c$ and possible biases. This process is fully data-free: both image generation and reasoning depend only on the model's behavior. For bias reasoning, we generate 50 images classified as the target class using DEXTER over the course of up to 5,000 optimization steps, employing a `multi-word` prompting strategy. These images are then captioned using ChatGPT-4o mini with the designated `Caption System Prompt` (detailed below). Subsequently, another instance of ChatGPT-4o mini analyzes these captions using the following `Report System Prompt` to extract key class attributes and identify potential biases in the model. To generate the captions and the reports with ChatGPT-4o mini we set a temperature of $0.2$ to mitigate hallucinations. The max tokens parameter is set to 0, allowing unrestricted response generation. The `top_p` is fixed at $1.0$, while both the `frequency_penalty` and the `presence_penalty` are set to $0.0$. Finally, the model generates $n = 1$ output.

We conducted multiple experiments on different visual classifiers:

- The reports presented in Fig. 4 and Appendix F.1 were obtained by analyzing a ResNet18 model trained on the *FairFaces* dataset.
- Figure 8, instead, provides a bias/not bias categorization for ViT, AlexNet, ResNet50, and RobustResNet50, all trained on the *ImageNet-1000* dataset, rather than full bias reasoning reports. Supplementary material contains the full bias reports for all the 30 classes (see Table 7) when analyzing RobustResNet50.

### Caption system prompt

```
You are an assistant specialized in analyzing pictures.
Your goal is to generate the most accurate and detailed description.
You will be given a picture and you will have to describe it inserting all the key attributes that you find that
could be inherent.
Keep the description short (one sentence) while including all the key attributes.

- You have to return only the description of the image. Don't return other information.
- Don't describe the image in a poetic way, just describe the visual elements.
- Don't include any information about the image that is not visible.
- Don't include any information about the image that is not relevant to the description.
- If the subject is a person describe the ethnicity features and age features.
- Keep the description short (one sentence) while including all the key attributes.

Examples:
Input image:  [IMG]
Input text:  Describe the image
Output:  [CAPTION]
```

### Report system prompt

```
Context:  This task involves analyzing images generated through a process of Class Activation Maximization (CAM).
Using a deep neural network trained on a specific dataset, the CAM approach identifies visual patterns that
strongly activate a specific class label in the network. To enhance this analysis, we use a trainable soft prompt
integrated with a Stable Diffusion model. The soft prompt is optimized to condition Stable Diffusion to create
images that maximize the activation of a chosen class. For each generated image, if the predicted class aligns
with the target class, a caption describing the image is stored.

Analysis:  Given a trainable soft prompt and a Stable diffusion model. We optimize the soft prompt to
conditionate stable diffusion to generate pictures that maximize the prediction of a class. If the class
predicted is the same of the class we want to maximize, than we store a caption of that picture.

Task:  Given a list of sentences that you will receive, you have to reason on the key attributes that the model
uses to maximize a specific class. Than you've to do a report with the explanation of your reasoning. The
report must end saying if the class is biased or not. If you reason that the class is biased you must explain why
otherwise you must explain why the class is not biased.

To write the report follow this structure:
- Title:  ### Report on Class Activation Maximization for the Class 'CLASS_NAME'
- Hypothesis on Model Bias

the class to analyze will be given by the user with the list of sentences. The shape of the user prompt will be
the following:

User:  'Class:  [CLASS] - Captions:  [LIST OF SENTENCES]'
```

## F.1 Gender Bias Reports

In this section, we report DEXTER's reports obtained by analyzing the ResNet18 classifier trained on the *FairFaces* dataset. As described in Section 4.2.3, we conducted two training procedures for this visual classifier. In the first procedure, we injected a gender bias across the classes, whereas in the second, we employed a balanced training set.

---

**Report for class 0 (20-29 years) of a biased classifier (overrepresentation of men)**

```
The analysis suggests that the class '20 to 29 years old people' may exhibit some bias.  While the model captures
a range of ethnicities and genders, the emphasis on certain skin tones, facial features, and expressions could
lead to a skewed representation of what is considered 'youthful'.
- Skin Tone Bias:  The frequent mention of light brown and medium brown skin tones may indicate a bias towards
these tones, potentially neglecting a broader spectrum of skin colors that also represent youth.
- Facial Feature Bias:  The focus on specific facial features that are often idealized in media could lead to a
narrow definition of youth, which may not encompass the full diversity of young adults.
- Gender Representation:  The slight focus on the overrepresentation of male subjects could suggest a bias in the
training data, which may not equally represent young women.
In conclusion, while the model successfully identifies attributes associated with the class '20 to 29 years old
people,' it is essential to recognize the potential biases in representation.  The model's predictions may not
fully encompass the diversity of this age group, indicating a need for more balanced training data to ensure
equitable representation across all demographics.
```

---

**Report for class 0 (20-29 years) of a balanced classifier (equal representation of men and women)**

```
Based on the analysis of the model, it does not appear to exhibit significant bias in identifying the class
of 20 to 29 years old individuals.  The diversity in hair and eye color, skin tone, and gender representation
suggests that the model is capable of recognizing a broad spectrum of individuals within this age group.  The
focus on well-defined facial features and neutral expressions aligns with common perceptions of young adults,
and the inclusion of various ethnic backgrounds indicates an inclusive approach.  In conclusion, the class of 20
to 29 years old people, as represented by the model, does not seem to be biased.  The attributes identified are
consistent with a diverse and inclusive representation of young adults.
```

---

**Report for class 1 (50-59 years) of a biased classifier (overrepresentation of women)**

```
Based on the analysis of the model, it can be concluded that the class '50 to 59 years old people' is biased.  The
bias stems from several factors:
- Representation of Age:  The model predominantly features youthful characteristics, which may not accurately
reflect the physical appearance of individuals aged 50 to 59.  This suggests that the model may be influenced by
societal standards that equate youthfulness with attractiveness, leading to a misrepresentation of the target age
group.
- Ethnic and Gender Bias:  The emphasis on lighter skin tones and the predominance of female subjects indicate
a lack of diversity.  This could perpetuate stereotypes and fail to represent the true diversity of individuals
within the 50 to 59 age range.
- Societal Stereotypes:  The model's reliance on neutral or serious expressions may reflect societal stereotypes
about older individuals, which can further skew the representation of this age group.
In conclusion, while the model attempts to maximize the activation of the class '50 to 59 years old people,' it
does so through a lens that is biased towards certain physical characteristics, expressions, and demographic
representations.  This highlights the importance of ensuring diverse and representative training data to mitigate
bias in machine learning models.
```

---

**Report for class 1 (50-59 years) of a balanced classifier (equal representation of men and women)**

```
Based on the analysis, it appears that the class '50 to 59 years old people' is not inherently biased.  The model
successfully captures a range of attributes that reflect the diversity of individuals within this age group.
However, there are potential biases in the representation of expressions and the predominance of serious or
neutral faces, which may not fully encompass the emotional spectrum of individuals aged 50 to 59.  Moreover, the
model's training data could influence its perception of age, potentially leading to stereotypes about how older
individuals should look or behave.  For instance, the emphasis on serious expressions may inadvertently reinforce
the stereotype that older adults are less expressive or joyful.  In conclusion, while the class itself is not
biased, the representation of certain attributes, particularly expressions and emotional portrayals, may benefit
from a broader range of depictions to avoid reinforcing stereotypes associated with aging.
```

## F.2 Bias Report Quality Assessment Metrics

In this section, we provide additional explanations for the metrics used to evaluate the quality of DEXTER's bias reports, as shown in Table 2. Following Liu et al. [20], we use the G-eval consistency metric, which involves prompting a large language model (LLM) to assess a text's factualness, scoring each report on a scale from 1 to 5. Since the original prompts for these metrics were designed for summarization, we create the following `G-eval Consistency Prompt` to better align with our task.

Following the G-eval format, we use the *Mean Opinion Score* of LLM ($MOS_{LLM}$) metric. This metric uses the $MOS_{LLM}$ prompt asking the LLM to rate between 1 and 5 the generated reports according to how well the report accurately describes the existence of bias in the classifier. Additionally, the $MOS_{humans}$ score represents the average rating given by participants in the user study.

For both the G-eval and $MOS_{LLM}$ configurations, we adopt the same settings as Liu et al. [20]. Specifically, we use `gpt-4-0613` with a temperature of 2, `max_tokens = 0`, `top_p = 1.0`, `frequency_penalty = 0`, and `presence_penalty = 0`.

To enhance response diversity, we generate $n = 20$ outputs. Furthermore, in both G-eval and MOS LLM system prompts, we provide the same evaluation instructions that were given to human evaluators during the user study, also following the best practices to reduce hallucinations provided by [4].

To assess whether the LLM's mean-opinion scores are statistically indistinguishable from those assigned by human annotators, we applied an independent-samples two one-sided equivalence test (TOST). The test compared the distribution of $MOS_{LLM}$ to that of $MOS_{humans}$ obtaining a $p \ll 0.05$. Thus, within the predefined equivalence bounds ($\pm 0.5$ for a Likert scale), the LLM's ratings can be considered statistically equivalent to human judgments, indicating strong agreement between the two sets of scores.

---

**G-eval Consistency system prompt**

```
**Instructions for Evaluation**:  In this task, we trained a visual classifier twice:
1.  First, using a dataset with gender-biased images.
2.  Second, using a balanced dataset.

Your task is to evaluate how effectively a given textual description identifies the presence or absence of gender
bias in the visual classifier:
- **Visual Classifier**:  A machine learning model trained to distinguish between two classes (e.g., Class 1:
"20-29 years old people" vs.  Class 2:  "50-59 years old people").

Please make sure you read and understand these instructions carefully.  Please keep this document open while
reviewing, and refer to it as needed.

Evaluation Criteria:

Consistency (1-5) - the factual alignment between the question and the description.  A factually consistent
description contains statements that are entailed by the question.  Annotators were also asked to penalize
descriptions that contained hallucinated facts.

Evaluation Steps:

1.  Read the question carefully and identify the main topic and key points.
2.  Read the description and compare it to the question.  Check if the description covers the main topic and key
points of the question, and if it presents them in a clear and logical order.
3.  Assign a score for consistency on a scale of 1 to 5, where 1 is "Bad Description" and 5 is "Excellent
Description" based on the Evaluation Criteria.

Example:

Question:

{{Question}}

Description:

{{Description}}

Evaluation Form (scores ONLY):

- Consistency (1-5):
```

## $MOS_{LLM}$ system prompt

**Instructions for Evaluation**: In this task, we trained a visual classifier twice:
1. First, using a dataset with gender-biased images.
2. Second, using a balanced dataset.

Your task is to evaluate how effectively a given textual description identifies the presence or absence of gender bias in the visual classifier:
- **Visual Classifier**: A machine learning model trained to distinguish between two classes (e.g., Class 1: "20-29 years old people" vs. Class 2: "50-59 years old people").

Please make sure you read and understand these instructions carefully. Please keep this document open while reviewing, and refer to it as needed.

Evaluation Criteria:

Mean Opinion score (1-5) - Assign a rating to the description based on its effectiveness:

- 1: Bad description (fails to identify bias or provide relevant details).
- 2: Poor description (some effort to address bias, but lacks clarity or completeness).
- 3: Neutral (adequate but not insightful; partially addresses bias).
- 4: Good description (clear and mostly thorough in addressing bias).
- 5: Excellent description (comprehensive, clear, and detailed in identifying bias).

Evaluation Steps:

1. Read the question carefully and identify the main topic and key points.
2. Read the description and compare it to the question. Check if the description covers the main topic and key points of the question, and if it presents them in a clear and logical order.
3. Assign a score for bias identification on a scale of 1 to 5, where 1 is "Bad Description" and 5 is "Excellent Description" based on the Evaluation Criteria.

Example:

Question:

{{Question}}
Description:

{{Description}}
Evaluation Form (scores ONLY):

- Rating (1-5):

### F.3 Bias identification of multiple vision classifiers

In Fig. 8 we report an example of disparity analysis across ViT, AlexNet, ResNet50, and Robus-tResNet50 for multiple ImageNet classes. By revealing class-specific biases and failure patterns, DEXTER helps identify where each model struggles and can guide data collection efforts when biases are consistently observed across classifiers. To select key neural features for these classifiers, we ranked penultimate-layer neurons by their weights to each class and selected the top 5, mirroring the Salient ImageNet method but without any training data.

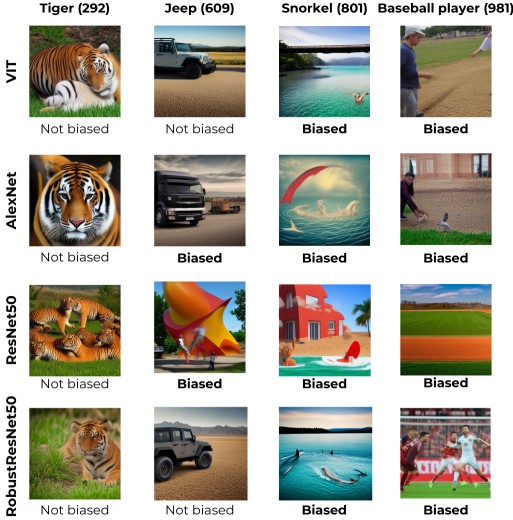

Figure 8: **Bias analysis across classifiers and SalientImageNet classes**. Each cell shows DEXTER-generated visual explanations and bias assessments ("Biased" or "Not Biased"). While Lion is unbiased across models, Jeep is biased in ResNet50 but not in its robust version. Baseball Player remains biased in all models, suggesting dataset-level bias.

# G  Additional details on Ablation Study

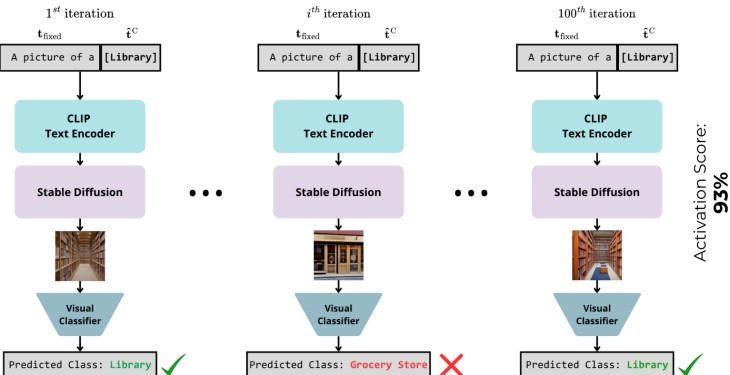

Figure 9: Procedure to compute the percentage of activations.

In this section, we provide further details on how we compute the activation score reported in Table 3 and Table 4. As shown in Figure 9, once the word(s) are obtained at the end of the optimization process, we prompt Stable Diffusion to generate 100 distinct images using a fixed prompt $\mathbf{t}_{\text{fixed}}$ (from the optimization stage) concatenated with the discovered word(s) $\hat{\mathbf{t}}^{\text{C}}$. We then sum all inference steps in which the visual classifier predicts the target class. Figure 10 illustrates examples of the different text-prompt strategies, described in Table 3, alongside their corresponding activation scores for the class Tiger and the class Snorkel.

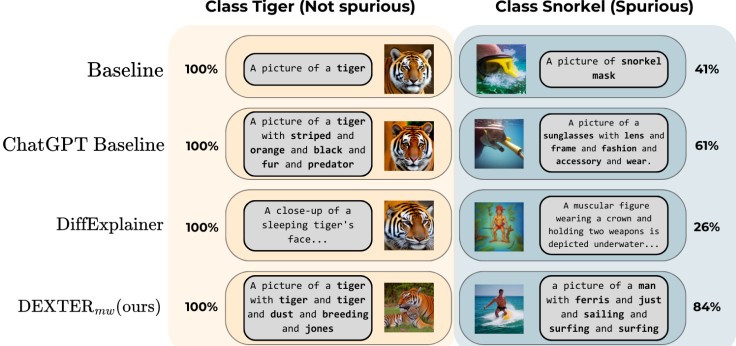

Figure 10: Examples of corresponding text prompts and generated images for class activation maximization. For a non-spurious class like tiger, all generated images easily activate the target class. However, in the case of the snorkel class, DEXTER is able to generate significantly more images that maximize the activation, and exposes the other features the model pays attention to.

## H  User Study Details

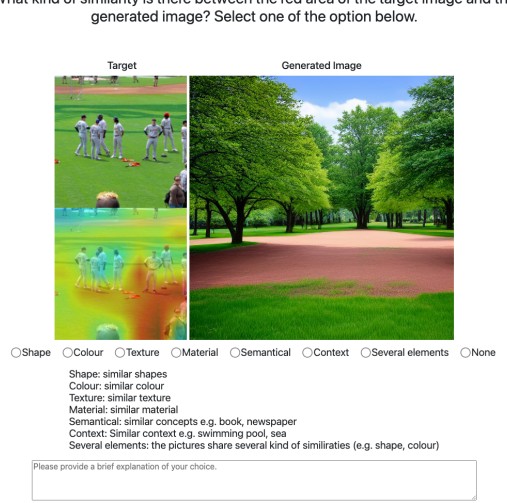

Figure 11: Example of a question from the first section of the user study.

Each user study consisted of two parts. The first part aimed to assess the ability of DEXTER to visually highlight the focus of the target classifier for a specific neural feature, while the second part evaluated the accuracy of the textual bias reports generated by DEXTER.

In the first part of the study, each participant was presented with 15 randomly images (explaining specific feature in SalientImageNet) drawn from the outputs of DiffExplainer and DEXTER. An example is shown in Fig. 11. Participants were asked to assess whether the generated image accurately represented the attended region of the real image based on attributes such as color, shape, texture, material, context, or semantic similarity. Additionally, two extra options were provided: *"Several Elements"* for cases where multiple similarity criteria were met, and *"None"* for instances where none of the criteria applied.

The second part of the study aimed to evaluate the ability of DiffExplainer reports to detect the presence of bias in the target classifier. The target classifier was trained both with and without gender bias, and participants were presented with all four reports generated by DEXTER. Examples of questions for class 0 (20–29 years old) are provided in Fig. 12 and Fig. 13, corresponding to the models trained with and without the injected gender bias, respectively. Participants were asked to rate, on a scale from 1 to 5, the extent to which the generated report reflected the presence or absence of bias.

The user study was conducted with 100 distinct MTurk workers, with an average completion time of 10.32 minutes. Each feature explanation in the first part received an average of 10.38 responses for DiffExplainer and 10.82 for DEXTER. To minimize response variability, participants were required to provide a textual justification for their answers in both sections of the questionnaire before submission. The compensation for each worker was $0.50.

## I  Diffusion and LLM's Hallucination evaluation

### I.1  Randomness in the diffusion generation process

To assess the stability of Stable Diffusion, we performed intra-class evaluation using DEXTER's final prompt. Across three independent runs (100 images each), we measured activation scores with the target model. If the image generation is unstable, scores will vary significantly. Instead, our consistent results indicate stability. We report mean and standard deviation for 8 randomly selected classes (4 core, 4 spurious) across three independent runs (Tab. 10).

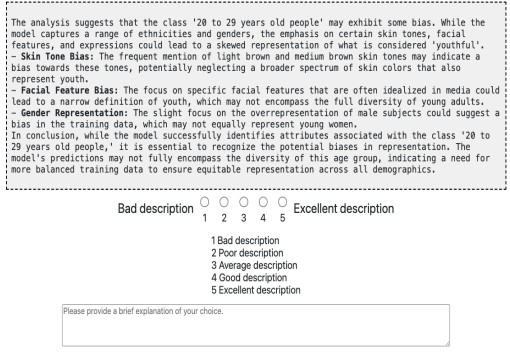

Figure 12: Example of a question from the second section of the user study (biased).

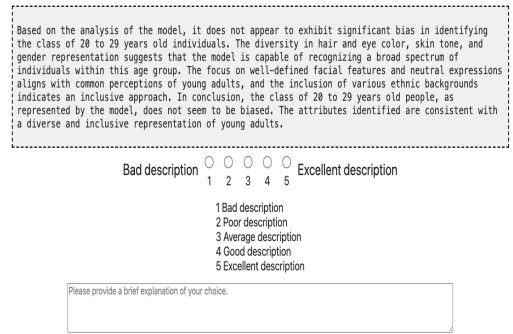

Figure 13: Example of a question from the second section of the user study (not biased).

Table 10: **Intra-class evaluation of randomness in Stable Diffusion generation process.**

| Type | Class | Avg |
|------|-------|-----|
| Spurious | 890 | $55.33 \pm 2.49$ |
| | 795 | $79.00 \pm 1.63$ |
| | 655 | $96.66 \pm 0.94$ |
| | 706 | $100.0 \pm 0.00$ |
| Core | 291 | $100.0 \pm 0.00$ |
| | 486 | $95.66 \pm 1.24$ |
| | 514 | $86.66 \pm 1.24$ |
| | 624 | $95.66 \pm 1.88$ |

## I.2 Bias propagation

To verify that DEXTER's outputs align with the classifier—rather than reflecting biases from BERT or Stable Diffusion—we ran a robustness test. We manually injected spurious cues into prompt initialization (e.g., replacing the starting auxiliary pseudo-target with *lion* for the class *tiger*) to simulate strong upstream bias. Then, for each class in SalientImageNet subset of our paper (split into "core" and "spurious" categories), we generated 100 images and measured how often the classifier activated the correct class. We report average classifier activation scores (as defined in Tab. 3 of the paper) for spurious, core classes, and their average. These results in Tab. 11 show that even with adversarially biased prompts, DEXTER recovers class-relevant visual patterns, aligning with the

classifier. Classifier-driven optimization helps correct upstream bias and grounds both visual and textual outputs in model behavior.

Table 11: **Comparison of Spurious, Core, and Average scores under different bias conditions.**

| Condition | Spurious | Core | Avg |
|---|---|---|---|
| Injected Bias | 65.3 | 88.4 | 76.8 |
| No Bias | 63.0 | 87.6 | 75.4 |

We also evaluated how DEXTER compensates for bias propagation. Specifically, we recorded the sequence of words selected as pseudotargets during optimization. This was done for both the biased setting and the standard setting, where the initial pseudotarget is chosen randomly. As shown in the Table 12, in both cases DEXTER consistently converges toward the most meaningful word, with the selection path gradually shifting from a random concept to the intended target. This trajectory suggests that the optimization process does not simply lock onto the first spurious or core concept it encounters, but instead explores a range of semantically plausible candidates before progressively refining the prompt toward a stable and interpretable solution, indicating a non-greedy and convergent behavior rather than an early commitment driven by the mask pseudo-label loss. Convergence analysis through keyword trajectory during the DEXTER optimization process. The Table 12 illustrates how DEXTER transitions from initially random concepts to progressively more semantically coherent and domain-specific words, demonstrating convergence toward a stable explanation.

### I.3 Analysis of LLM hallucination impact

Given that large language models can generate statements influenced by their prior knowledge rather than by factual evidence, we systematically assess the fidelity of our reports and how much they are aligned with the visual classifier decision-making process. We aim to ensure that each report highlights genuine visual patterns exploited by the classifier rather than hallucinated elements. Concretely, for each of the 30 SalientImageNet classes in our experimental setup (Tab. 7), we verify whether the single most salient visual cue extracted from the corresponding DEXTER report is truly *grounded*. Given a class $c$ we form

> BASELINE PROMPT: *"a picture of a [c]"*,
> CUE PROMPT: *"a picture of a [c] with [cue]$_0$ ... and '[cue]$_n$'*,

where $n$ is the total number of cues obtained automatically from the generated reports via a GPT–4o mini class-cues extractor. Then, we generate 100 images per prompt with Stable–Diffusion and measure the *Activation Score (AS)* on the frozen *RobustResNet50* used throughout the paper (4.2.4). A cue is **grounded** when $AS_{cue} > AS_{baseline}$; otherwise it is neutral or wrong.

**Statistical significance & confidence interval.** For each class we measure $\Delta_i = AS_{cue,i} - AS_{baseline,i}$. Across the 30 classes we obtain

$$\bar{\Delta} = 16.33 \text{ pp}, \qquad s_\Delta = 23.84 \text{ pp}.$$

The two-sided 95% confidence interval is

$$\bar{\Delta} \pm t_{0.975,29} \frac{s_\Delta}{\sqrt{30}} = [\,7.43,\ 25.24\,] \text{ pp}.$$

Both a paired $t$-test ($t(29) = 3.79$, $p = 7.8 \times 10^{-4}$) and a Wilcoxon signed-rank test ($W = 19$, $p = 2.9 \times 10^{-4}$) confirm that the improvement is statistically significant.

**Aggregate results.** Table 13 contrasts the mean Activation Scores for the two prompts. Adding the report cue raises the mean score from **64.73%** to **81.06%** (+16.33 pp).

- **20/30 classes** (67%) improve (peak +81 pp for *miniskirt*), confirming that the cues capture genuinely discriminative evidence.
- **7 classes** already achieve 100 % with the baseline prompt; the cue therefore leaves performance unchanged.

Table 12: **Word-wise convergence path for biased vs unbiased initialized prompt.**

| Grocery Store (582) | | Patio (706) | |
|---|---|---|---|
| **Biased** | **Not Biased** | **Biased** | **Not Biased** |
| motel | fan | restaurant | head |
| library | coffin | school | book |
| wedding | factory | house | man |
| party | sale | pool | the |
| restaurant | factory | residence | republic |
| party | distillery | | tree |
| wedding | market | | window |
| motel | supermarket | | house |
| bakery | | | courtyard |
| supermarket | | | terrace |
| | | | patio |

Table 13: **Activation Scores averaged over 30 classes ($\uparrow$/$\downarrow$: cue better/worse than baseline).**

| Prompt | Mean AS (%) | $\Delta$ | Class split |
|---|---|---|---|
| Baseline | 64.73 | — | |
| Cue (report) | 81.06 | +16.33 | 20$\uparrow$  3$\downarrow$  7 = 30 |

- **3 classes** degrade:

| | |
|---|---|
| *bubble* | $\Delta = -5$ pp; the cue is correct but pertains only to a subset of bubble instances. |
| *tanks* | $\Delta = -1$ pp ($100\% \to 99\%$); a negligible loss. |
| *rifle* | $\Delta = -17$ pp; the cue *"individuals in dark clothing"* marking the sole clear failure. |

**Overall assessment.**   DEXTER explanations are therefore *strongly grounded*: in 27 out of 30 classes the cue is beneficial or neutral, and only one class (*rifle*) exhibits evidence of hallucination. In the two minor degradations (*bubble*,*tanks*) the reports still isolate genuine—though partial—visual cues.

**Evaluation Implications.**   The consistent increase in classifier confidence when report-derived cues are injected into the prompt demonstrates that DEXTER's explanations faithfully align with the model's decision boundary, rather than reflecting spurious or hallucinatory artifacts. Together with the text-based metrics reported in Table 2 for the FairFaces dataset (*G-Eval consistency*, *STS*, *MOS*), these findings confirm that DEXTER delivers genuinely grounded insights into the visual classifier's decision-making process.

**Pipeline Robustness.**   Furthermore, this evaluation implicitly validates the collaborative operation of all pretrained components (BERT, CLIP, Stable Diffusion, VLM) in our optimization pipeline. The target visual classifier guides the prompt optimization. BERT selects discriminative keywords, the diffusion model generates contextually relevant images, and the captioning VLM produces faithful captions. As a result, the end-to-end process avoids propagating biases or hallucinations from intermediate models and yields explanations that are solidly grounded in the classifier's behavior.

## I.4   Baseline prompts vs Cue prompts

This section, referring to Sec. I.3, reports the system prompt used to extract the relevant visual cues from DEXTER's textual explanations. Furthermore, Table 14 compares class-wise activation scores produced by the baseline prompt ``A picture of a [CLASS]'' and by the CUE-enriched prompt. Large gains appear for classes that were poorly activated in the baseline (e.g., space bar improves from $6 \to 57$, hockey puck from $0 \to 79$). The results demonstrate that adding concise semantic cues markedly improves class-specific guidance and reduces hallucinations during image generation.

## Report's cues extractor system prompt

```
You are an assistant that reads a bias-analysis report about an ImageNet class and extracts concrete visual cues
that the report claims are important for that class.
Your task
1.  Identify up to 5 key visual phrases (2-5 words each) that:

              • are explicitly mentioned in the report;

              • describe tangible elements that can appear in an image (objects, attributes, background, actions);

              • are likely to trigger the classifier according to the report.

2.  Return your answer in JSON with two fields:
'''json  "key-phrases":  ["phrase1", "phrase2", ...], "full-prompt":  "a picture of a <CLASS> with "phrase1" and
"phrase2" and ..."
```

Table 14: **Activation maximization scores for each class: comparison between the baseline prompt *"A picture of a [CLASS]"* and the CUE prompt obtained from the DEXTER's reports.**

| Class | Baseline prompt | AS | CUE prompt | AS |
|---|---|---|---|---|
| volleyball | A picture of a volleyball | 66 | A picture of a volleyball with young women playing and mixed-gender teams and variety of scenarios | 78 |
| space bar | A picture of a space bar | 6 | A picture of a space bar grid layouts with framed artworks and black-and-white imagery | 57 |
| umbrella | A picture of an umbrella | 95 | A picture of an umbrella with vibrant outdoor scenes and garden scenes | 100 |
| baseball player | A picture of a baseball player | 98 | A picture of a baseball player with female athletes and softball context | 100 |
| bubble | A picture of a bubble | 100 | A picture of a bubble with whimsical themes | 95 |
| balance beam | A picture of a balance beam | 0 | A picture of a balance beam with athletic attire and physical activities and indoor training environments and group dynamics and artistic performances | 27 |
| cowboy boot | A picture of a cowboy boot | 83 | A picture of a cowboy boot with various types of boots | 96 |
| patio | A picture of a patio | 100 | A picture of a patio with modern architecture and expansive outdoor spaces and affluent homes and specific landscaping styles | 100 |
| tank | A picture of a tank | 100 | A picture of a tank with camouflage and military environment and armored vehicle | 99 |
| dark glasses | A picture of dark glasses | 8 | A picture of dark glasses with facial hair and an older man | 20 |
| daisy | A picture of a daisy | 100 | A picture of a daisy with vibrant colors and natural contexts and red daisies | 100 |
| howler monkey | A picture of a howler monkey | 0 | A picture of a howler monkey with lush, green environments and natural habitat | 32 |
| tiger | A picture of a tiger | 100 | A picture of a tiger with striped fur and natural habitat and majestic posture | 100 |
| library | A picture of a library | 76 | A picture of a library with books and organized indoor area and spacious room with furniture and people | 97 |
| seat belt | A picture of a seat belt | 56 | A picture of a seat belt with human subjects | 73 |
| rifle | A picture of a rifle | 85 | A picture of a rifle with individuals in dark clothing and jackets and sunglasses and aiming | 68 |
| grocery store | A picture of a grocery store | 84 | A picture of a grocery store with diversity of food items and presence of people and colorful displays | 95 |
| snorkel | A picture of a snorkel mask | 41 | A picture of a snorkel mask with young individuals and sharks | 94 |
| dogsled | A picture of a dogsled | 98 | A picture of a dogsled with snowy landscapes and human-animal interaction and specific dog breeds | 100 |
| magnetic compass | A picture of a magnetic compass | 6 | A picture of a magnetic compass with silver compass pendant and craftsmanship and design | 26 |
| horizontal bar | A picture of a horizontal bar | 6 | A picture of a horizontal bar in sports and athletic environment | 8 |
| ski | A picture of a ski | 56 | A picture of a ski with young individuals with ski attire | 100 |
| miniskirt | A picture of a miniskirt | 19 | A picture of a miniskirt with women wearing miniskirts | 100 |
| lion | A picture of a lion | 100 | A picture of a lion with distinctive physical traits and social behavior and natural habitat and human interactions | 100 |
| hockey puck | A picture of a puck | 0 | A picture of a puck with ice hockey gameplay and player confrontations | 79 |
| swimming cap | A picture of a swimming cap | 92 | A picture of a swimming cap with children in joyful scenarios with adults and certain hair types and smiling expressions | 99 |
| bulletproof vest | A picture of a bulletproof vest | 67 | A picture of a bulletproof vest with casual attire | 89 |
| cello | A picture of a cello | 100 | A picture of a cello with musical instrument | 100 |
| golf ball | A picture of a golf ball | 100 | A picture of a golf ball with spherical objects and white color | 100 |
| jeep | A picture of a jeep | 100 | A picture of a jeep, landrover with off-road capability | 100 |
| **Mean** | — | **64.73** | — | **81.06** |


Table 15: Assets used and licence information.

| Asset | Type | License | Github / URL | Citation/Reference |
|---|---|---|---|---|
| SalientImagenet | data | non-commercial research | singlasahil14/salient_imagenet | [40] |
| Waterbirds | data | non-commercial research | kohpangwei/group_DRO | [35] |
| CelebA | data | non-commercial research | https://mmlab.ie.cuhk.edu.hk/projects/CelebA.html | [21] |
| FairFaces | data | CC BY 4.0 | joojs/fairface | [14] |
| BERT | Pretrained Model | Apache license 2.0 | https://huggingface.co/google-bert/bert-base-uncased | [5] |
| CLIP | Pretrained Model | MIT license | openai/CLIP | [31] |
| Stable Diffusion 1.5 | Pretrained Model | CreativeML Open RAIL-M | https://huggingface.co/stable-diffusion-v1-5/stable-diffusion-v1-5 | [34] |
| GPT-4o | LLM | API-based use under OpenAI terms | https://platform.openai.com/docs/overview | OpenAI |

    Guidelines:

    - The answer NA means that the paper does not use existing assets.
    - The authors should cite the original paper that produced the code package or dataset.
    - The authors should state which version of the asset is used and, if possible, include a URL.
    - The name of the license (e.g., CC-BY 4.0) should be included for each asset.
    - For scraped data from a particular source (e.g., website), the copyright and terms of service of that source should be provided.
    - If assets are released, the license, copyright information, and terms of use in the package should be provided. For popular datasets, paperswithcode.com/datasets has curated licenses for some datasets. Their licensing guide can help determine the license of a dataset.
    - For existing datasets that are re-packaged, both the original license and the license of the derived asset (if it has changed) should be provided.
    - If this information is not available online, the authors are encouraged to reach out to the asset's creators.

13. **New assets**

    Question: Are new assets introduced in the paper well documented and is the documentation provided alongside the assets?

    Answer: [NA]

    Justification: While the paper introduces DEXTER as a novel framework and reports experimental results using standard datasets, it does not release new datasets or models as our proposed method aims to explain a pretrained model (classifier) using other pretrained frozen models.

    Guidelines:

    - The answer NA means that the paper does not release new assets.
    - Researchers should communicate the details of the dataset/code/model as part of their submissions via structured templates. This includes details about training, license, limitations, etc.
    - The paper should discuss whether and how consent was obtained from people whose asset is used.
    - At submission time, remember to anonymize your assets (if applicable). You can either create an anonymized URL or include an anonymized zip file.

14. **Crowdsourcing and research with human subjects**

    Question: For crowdsourcing experiments and research with human subjects, does the paper include the full text of instructions given to participants and screenshots, if applicable, as well as details about compensation (if any)?

Answer: [Yes]

Justification: All information about the user study conducted in this work is provided in both the main paper and the appendix, including the results and the questionnaire administered to human evaluators. The appendix contains the full text of the questionnaire as well as details about participant compensation for each HIT.

Guidelines:

- The answer NA means that the paper does not involve crowdsourcing nor research with human subjects.
- Including this information in the supplemental material is fine, but if the main contribution of the paper involves human subjects, then as much detail as possible should be included in the main paper.
- According to the NeurIPS Code of Ethics, workers involved in data collection, curation, or other labor should be paid at least the minimum wage in the country of the data collector.

15. **Institutional review board (IRB) approvals or equivalent for research with human subjects**

Question: Does the paper describe potential risks incurred by study participants, whether such risks were disclosed to the subjects, and whether Institutional Review Board (IRB) approvals (or an equivalent approval/review based on the requirements of your country or institution) were obtained?

Answer: [NA]

Justification: The paper does not involve research with human subjects.

Guidelines:

- The answer NA means that the paper does not involve crowdsourcing nor research with human subjects.
- Depending on the country in which research is conducted, IRB approval (or equivalent) may be required for any human subjects research. If you obtained IRB approval, you should clearly state this in the paper.
- We recognize that the procedures for this may vary significantly between institutions and locations, and we expect authors to adhere to the NeurIPS Code of Ethics and the guidelines for their institution.
- For initial submissions, do not include any information that would break anonymity (if applicable), such as the institution conducting the review.

16. **Declaration of LLM usage**

Question: Does the paper describe the usage of LLMs if it is an important, original, or non-standard component of the core methods in this research? Note that if the LLM is used only for writing, editing, or formatting purposes and does not impact the core methodology, scientific rigorousness, or originality of the research, declaration is not required.

Answer: [Yes]

Justification: The paper explicitly describes the use of large language models (LLMs) as a central and original component of its methodology. In DEXTER, LLMs are used to generate textual prompts and provide human-interpretable explanations of visual classifier behavior.

Guidelines:

- The answer NA means that the core method development in this research does not involve LLMs as any important, original, or non-standard components.
- Please refer to our LLM policy (https://neurips.cc/Conferences/2025/LLM) for what should or should not be described.

