# OpenReview forum: "DEXTER: Diffusion-Guided EXplanations with TExtual Reasoning for Vision Models"
_NeurIPS.cc/2025/Conference — NeurIPS 2025 spotlight_

### Official Review · Reviewer_cbH4 · 2025-06-29

**Clarity:** 2
**Significance:** 3
**Originality:** 3
**Rating:** 5
**Confidence:** 4

**Summary:**

The paper proposes a framework for generating ​global, textual explanations​ of visual classifiers without requiring training data. It first optimizes text prompts which are applied to generate class-specific images that maximally activate a target classifier. Then it utilizes large language models to analyze synthetic images to produce natural language reports on model behavior and decision patterns. The framework can be operated in tasks like activation maximization, slice discovery for debiasing and bias explanation.

**Questions:**

I welcome the authors' comments on the concerns outlined in the weaknesses section of my review and their input to clarify any points I may have misunderstood.

**Ethical Concerns:**

["NO or VERY MINOR ethics concerns only"]

**Final Justification:**

I decide to maintain the rating 5 considering all the reviews and response from the authors. The main reasons are:
1. For my concerns of unclear model pipeline, the authors addressed it by clarifying with simple and straightforward steps, which also matches my understanding.
2. For the confusions of neuron subset selection, which also mentioned by other reviewers, the authors provides clear and convincing answers.
3. For other technical points mentioned by other reviewers, most of them are addressed. Some are left for further improvement or exploration in the future.
In view of all factors above, I have such a conclusion and decision.

**Limitations:**

yes

**Paper Formatting Concerns:**

No Paper Formatting Concerns

**Quality:**

3

**Strengths And Weaknesses:**

Strengths:
1. Novel residual-like mask pseudo-labels prediction to support the integrated differentiable process for optimizing text prompts. The design can make joint optimization possible rather than treat prompt generation and image synthetis separately.
2. Flexible application in various tasks. The method is applicable across tasks like bias detection, feature visualization, debiasing, which might be very useful in model performance improvement.
3. Abundant and reasonable emprical design and results. Following multiple applicable tasks, it prepares sufficient qualitative and quantitative results to support claim well.

Weaknesses:
1. Unorganized method illustration. From section 3.1-3.3, in fact there are a lot of steps with complicated notations, K, V, agg, act, .etc, which is a huge chaos. Then it is hard to follow. Even if there is an overall loss function in section 3.3, it only works after certain pseudo-label has been set. So, the order and dependancy among different steps are unclear. A clear and integrated algorithm pipeline might be helpful.
2. Insufficient details for some key steps. In section 3.3, for each pseudo-label $y_{i}$, its aggregated activation loss L_{agg, i} is initialized and optimized upon a subset of reference neurons $\mathcal{N}_{i}$. It is unclear how the subset is defined or changed across the optimization process.

---

> ### Author Rebuttal · Authors · 2025-07-30
>
> We appreciate the reviewer’s careful reading of our paper and the insightful feedback. Below, we respond to the main questions (Q), aiming to clarify and strengthen our contribution.
>
>
> **[Q1 - Unorganized method illustration]**
>
> We acknowledge that Sections 3.1–3.3 involve several interdependent steps and notations, which may appear complex at first read. To improve clarity, we will revise the paper to include:
> - A streamlined algorithm-style summary of the full pipeline;
> - An updated illustration (Figure 1) that explicitly shows the data flow across components;
> - A glossary clarifying key notations (e.g., M, K, V, agg, act).
> For clarity, here is a high-level overview of the pipeline:
> 1) Section 3.1 main paper
>    - We begin with a visual classifier that we want to investigate if it is biased on a class (e.g. "bee")
>    - We form the input for a BERT model as follows: [Soft Prompt] + ["A picture of a"] + [Masked Tokens]
>    - The BERT model is tasked with predicting the masked tokens, in the initial steps, it will likely predict the most common words it is pretrained on (e.g. "man")
>    - The predicted BERT token is translated into the corresponding CLIP token using the Gumbel Softmax and the translation matrix defined in Sec. 3.1 as $M$.
> 2) Section 3.2 main paper
>    - This token is then encoded with the sentence "A picture of a" using a CLIP Text Encoder to form the encoded sentence "A picture of a [PREDICTED CLIP TOKEN]"
>    - This encoded sentence is used to text-condition a Stable Diffusion model and generate a corresponding image
>    - The generated image is then classified using the investigated visual classifier, and a cross-entropy (CE) loss is obtained.
> 3) Section 3.3 main paper
>    - To update the soft prompt we sum the cross-entropy with the masked loss (w.r.t. the pseudo-label obtained in the previous iteration as described in 9-10) and backpropagate across the full pipeline
>    - We update the dictionary where we store the historical CE losses for each token
>    - To supplement the weak training signal, on successive iterations, we use the token having the lowest mean historical loss as a pseudo-label to supervise the future predicted BERT tokens with a masked loss. This allows us to boost the training signal when updating the soft prompt
>    - Over many iterations, the soft prompt will condition the BERT encoder to predict the most relevant word to maximize the model's prediction of the investigated class. In the Figure 1 example, the model's most relevant word is "flower" when maximizing the class "bee". This indicates that the model is inherently biased on the "bee" class.
>
> These clarifications and structural improvements will be included in the final version.
>
> **[Q2 - Insufficient details for some key steps]**
>
> We thank the reviewer for the observation. In the case of RobustResNet50, neural feature selection is guided by pseudo-ground truth from Salient ImageNet, which is precisely why we chose this model for our in-depth evaluation. For other classifiers lacking such annotations, the subset of reference neurons used to compute the aggregated activation loss $\mathcal{L}_{\text{agg}, i}$ is defined statically before optimization and remains fixed throughout. Specifically, we select the top-k neurons $\{n_1, …, n_k\}$ in the penultimate layer with the highest connection weights to the target class output. This ensures that optimization is consistently influenced by the most class-relevant neurons. We will clarify this detail more explicitly in the revised version.  We will clarify this point more explicitly in the revised version.

---

> > ### Comment · Reviewer_cbH4 · 2025-08-03
> >
> > Thanks for the response from authors. It has addressed most of my confusions and concerns. I hope the clarification can be well integrated into revised version of the paper.

---

### Official Review · Reviewer_xK8N · 2025-07-01

**Clarity:** 3
**Significance:** 3
**Originality:** 3
**Rating:** 5
**Confidence:** 4

**Summary:**

This paper introduces DEXTER, a framework for generating images that strongly activate targeted neurons in a given classifier.
Its key components are :
- a frozen Language Model (LM), pre-trained to perform Masked Language Modeling (MLM), which is used to fill in masked tokens in a template prompt, conditioned by a soft prompt vector;
- a frozen CLIP text encoder that encodes the hard prompt completed by the LM;
- a Stable Diffusion model that generates images conditioned by the embedding produced by the CLIP text encoder.

The LM’s soft prompt vector is the only trainable component in the pipeline.

The words filled in by the LM can be used to identify spurious correlations learned by the classifier or for interpreting neural features. A bias report for the classifier can also be obtained by using a VLM to caption a set of generated images and then prompting the same VLM to analyse all the captions.

**Questions:**

Questions

1. Are the activation scores in Tables 3 and 4 the mean activations for the correct class across multiple images? If so, could the authors also provide the corresponding standard deviations?

2. In Appendix C how are the top-k words from Table 7 selected? Is the pipeline run multiple times and the top word of each run is selected (the last pseudo-target of $\mathcal{L}_{mask}$), or do the author select the top-k words based on the activations produced in a single run?

3. For the words in Table 7, could the authors provide the scores used to rank them, as well as the frequency at which they were sampled in the optimization process? (this is related to question 2 and the discussion below)

While the method can be used to identify spurious correlations (SCs) learned by the classifier (the experiments on the Waterbirds and CelebA datasets), the diversity of the proposed words seems a bit low (Appendix Table 7), at least when compared to methods such as B2T[1], Lg[2] or WASP[3]. This is not a major issue, as the identification of SCs is not the main goal of DEXTER, unlike the previously mentioned works.

However, I was wondering if qualitative improvements could be made in this regard by simply selecting more words sampled from the MLM encoder in the optimization process. For example, by ranking the sampled words by the strength of the produced activations in the classifier and presenting them in this order, together with the number of times they were sampled from the MLM encoder. This information should be freely available and I think that it would make for an interesting addition to the Appendix.

It could also offer more insights on how the optimization process works - does it explore multiple valid SCs in this setup and then converges to one of them, or does it simply lock onto the first SC that it finds? A slight concern would be that the mask pseudo-label loss ($\mathcal{L}_{mask}$) might limit the exploration of potential SCs.

This type of analysis could also be done for the experiment in Appendix G.1, where the pseudo-label is initialized with the wrong class.

Suggestions:
- The results in Section 4.2 are a bit hard to parse - e.g. the caption of Table 3 doesn’t seem to offer any sort of hint as to what those numbers represent. I think the clarity of the presented results should be improved in this regard.
- There are a few typos that should be fixed  - $\mathcal{L}$mask, on page 9, lines 319 and 322.

[1] Kim et al. "Discovering and mitigating visual biases through keyword explanation." Proceedings of the IEEE/CVF Conference on Computer Vision and Pattern Recognition. 2024.

[2] Zhao et al. "Language-guided detection and mitigation of unknown dataset bias." arXiv preprint arXiv:2406.02889 (2024).

[3] Paduraru et al. “WASP: A Weight-Space Approach to Detecting Learned Spuriousness.” arXiv preprint arXiv:2410.18970 (2024).

**Ethical Concerns:**

["NO or VERY MINOR ethics concerns only"]

**Final Justification:**

DEXTER, the framework introduced in this work is flexible and validated for multiple tasks.

The authors have addressed the concerns of the reviewers in their rebuttal and clarified aspects related to the methodology and the presented results.

The authors compare DEXTER with DiffExplainer, a prior method that they built upon, highlighting the improvements and benefits brought by their novel additions.

The  hardware requirements of the framework are justified by the nature of the application - model explainability - and the authors have taken measures to reduce this limitation (e.g., using modules that are more computationally efficient).

Overall, the proposed procedure seems sound and effective. Integrating the reviewers’ feedback should improve the clarity of the paper.

**Limitations:**

The authors analysed the limitations of their method in Section 5.

**Paper Formatting Concerns:**

No formatting concerns were noticed in this paper.

**Quality:**

3

**Strengths And Weaknesses:**

Strengths:
1. The framework is modular - the models used in the pipeline can be replaced with newer or specialized ones for domain-specific classifiers;
2. It can perform bias or spurious correlation identification in a data-free manner - it does not require access to the training data of a classifier in order to identify spurious correlations that the classifier has learned;
3. The framework’s effectiveness is proven in multiple tasks (activation maximization, slice discovery, bias explanation).

Weaknesses:
1. As noted by the authors, the framework is rather demanding in terms of hardware resources because it needs to perform backpropagation through the MLM encoder, the CLIP text encoder used to condition a diffusion model, the diffusion model itself and the targeted classifier;
2. The initial objective (maximizing neural activations) is difficult to optimize, most likely due to the length of the backpropagation chain. To address this issue, the authors resorted to an auxiliary loss $\mathcal{L}_{mask}$ added after the MLM encoder, which encourages the encoder to predict words that have historically led to higher activations in the classifier. Without this second loss term the results obtained in Table 4 are drastically lower for the multi-word prompt optimization;
3. The diffusion model can generate additional objects besides what is expressed in the textual prompt, thus inducing some uncertainty with regards to what actually leads to an increased activation for a neuron - a concept mentioned in the textual prompt or a different one that is additionally generated by the diffusion model. This matters in the slice discovery application, but not as much for the bias explanation one, where the generated images are separately captioned and the bias report is generated from these extended captions;
4. Each component in the pipeline limits the concepts that the framework can identify as important for the classifier (words that the MLM encoder tends to predict in the template prompt and are also in the vocabulary of CLIP’s text encoder, or concepts that the diffusion model can generate), but this can be addressed by upgrading any lacking component (assuming that an appropriate replacement exists)

---

> ### Author Rebuttal · Authors · 2025-07-31
>
> We thank the reviewer for the careful evaluation of our work and the thoughtful comments. In the following, we provide clarifications and further insights in response to the weaknesses (W) and questions (Q) raised.
>
> **[W1 – Computational Demand]**
>
> We appreciate the reviewer’s observation. As noted in the limitations section, DEXTER’s optimization involves multiple frozen models, which can be resource-intensive. To mitigate this, we adopt half-precision arithmetic and latent-consistent diffusion, which significantly reduce runtime. While the prompt optimization step takes ∼10 minutes per class, the final explanation generation is very efficient (∼15 seconds), making the framework practical for offline, global interpretability tasks.
>
> **[W2 – Optimization Difficulty of Activation Objective]**
>
> We agree that the activation maximization loss alone can lead to slow convergence. The auxiliary pseudo-label loss was introduced specifically to address this issue, and as shown in our ablation (Table 4), it substantially improves performance, especially in multi-word scenarios. We see this not as a workaround, but as a targeted mechanism to improve convergence and interpretability.
>
> **[W3 – Spurious Object Generation by Diffusion Model]**
>
> We acknowledge that diffusion models may generate elements beyond the text prompt. However, our optimization loop includes feedback from the classifier, which gradually narrows the generation space to align with the desired activation patterns. As a result, off-topic generations tend to be rare and are filtered out over time. This is particularly effective in slice discovery, where consistency across images is critical.
>
> **[W4 – Vocabulary and Generative Constraints]**
>
> We agree that the intersection of vocabularies and model capabilities can limit the expressiveness of the generated explanations. However, the modular nature of our framework allows for straightforward integration of improved components, such as larger MLMs or more expressive diffusion models. We see this as a promising direction for future enhancement.
>
> **[Q1 – Activation Scores in Tables 3 and 4]**
>
> Tables 3 and 4 report the mean activation scores across the 30 classes analyzed by DEXTER. For each class, we generated 100 images using the optimized prompt (e.g., for the class “ski,” 77 out of 100 generations activated the target neuron). The reported scores reflect the average number of successful activations per class.
> As requested, we also provide the corresponding standard deviations. These may appear high due to the variation in activation strength across different classes, some classes are consistently activated, while others only partially so. For instance, while some classes were activated in 100 out of 100 generations, others only in 20 out of 100. As a result, this large variation across classes naturally leads to a high overall standard deviation.
> As requested by the reviewer, below we report the scores from Tables 3 and 4, followed by their corresponding standard deviations:
>
> Tab. 3 main paper: Comparison between text-prompting strategies for maximizing the target class.
>
> |         | **Spurious**       | **Core**           | **Mean**           |
> |-|-|-|-|
> | **Baseline** | 43.06 ± 38.86      | 86.40 ± 23.83      | 64.73 ± 31.34      |
> | **ChatGPT**  | 41.20 ± 40.78      | 78.53 ± 34.02      | 59.87 ± 37.40      |
> | **Diffexplainer** | 33.20 ± 41.07 | 47.66 ± 44.80      | 39.83 ± 42.93      |
> | **DEXTER**   | **63.00 ± 31.20**  | **87.86 ± 15.14**  | **75.43 ± 23.17**  |
>
> Tab. 4 main paper: Ablation results for single-word and multi-word prompt optimization, with/without the auxiliary task $\mathcal{L}_{mask}$.
>
> |                             | **Spurious**       | **Core**           | **Mean**           |
> |-|-|-|-|
> |**DEXTER single mask**          | 11.13 ± 27.38      | 36.33 ± 38.45      | 23.73 ± 32.91      |
> | **DEXTER single mask + $\mathcal{L}_{mask}$**    | 34.00 ± 32.72      | 53.86 ± 44.64      | 43.93 ± 38.68      |
> | **DEXTER multi mask**           | 15.53 ± 27.93      | 08.13 ± 18.74      | 11.83 ± 23.33      |
> | **DEXTER multi mask + $\mathcal{L}_{mask}$**     | **63.00 ± 31.20**  | **87.86 ± 15.14**  | **75.43 ± 23.17**  |
>
> Importantly, across all conditions, DEXTER achieves both higher average activation scores and lower standard deviations compared to previous methods, indicating more robust and consistent performance. We will include this clarification and full results in the final version.
>
> **[Questions 2 and 3 - Selection of top-k words from Table 7]**
>
> The top-k words in Table 7 (Appendix C) were obtained by running the DEXTER pipeline multiple times (specifically k=4 timeless). In each run, we recorded the final (only one) pseudo-token selected by the masked language model at the end of the optimization. To encourage diversity across runs, we excluded previously selected words from the candidate pool before each new run.
> We fully agree that using ranked words, rather than only one,  would provide valuable insight into the optimization process and the identification of spurious correlations. Indeed, we adopted a similar ranking procedure in other parts of our framework (e.g., in the bias report and activation score evaluation), specifically:
>
> - We track all candidate prompts (e.g., “a picture of a [WORD] with [WORD]…”) generated during optimization.
> - For each, we compute an activation score by generating 100 images and measuring how many are classified as the target class.
> - The prompt with the highest activation score is selected as the final output.
>
> However, as suggested, we also evaluated how using sampling frequency ranking may affect slice discovery results. In particular, we computed results using word frequency–based ranking, evaluating performance with the top k = 5 and k = 10 most frequent words. We compare these results against our proposed sampling strategy and B2T for reference. As shown, our strategy with a single word yields superior performance.
>
> | Method          | F1-score Class 0 | F1-score Class 1 |
> |-|-|-|
> | B2T             | **0.99**             | 0.75             |
> | k = 10          | 0.98             | 0.67             |
> | k = 5           | 0.98             | 0.66             |
> | ours | **0.99**             | **0.76**             |
>
> We believe the above results can be explained with the fact that running multiple independent optimizations, each targeting a single word and discarding previously discovered ones, encourages exploration of diverse semantic directions. In contrast, a single optimization run using the top-k words tends to converge quickly and remains confined to a narrow neighborhood of semantically related tokens. This behavior limits the variety of the discovered words and, consequently, the coverage of distinct slices. Nonetheless, we agree that a more systematic analysis of the full distribution of sampled words, as well as the design of more efficient selection strategies, represents a promising direction for future work.
>
> Regarding the analysis on how DEXTER compensates for bias propagation in Appendix G.1, we recorded the sequence of words selected as pseudotargets during optimization. This was done for both the biased setting (i.e., under the same conditions as those reported in Appendix G.1) and the standard setting, where the initial pseudotarget is chosen randomly. As shown in the table below, in both cases DEXTER consistently converges toward the most meaningful word, with the selection path gradually shifting from a random concept to the intended target. This trajectory suggests that the optimization process does not simply lock onto the first spurious or core concept it encounters, but instead explores a range of semantically plausible candidates before progressively refining the prompt toward a stable and interpretable solution, indicating a non-greedy and convergent behavior rather than an early commitment driven by the mask pseudo-label loss.
> Convergence analysis through keyword trajectory during the DEXTER optimization process. The table illustrates how DEXTER transitions from initially random concepts to progressively more semantically coherent and domain-specific words, demonstrating convergence toward a stable explanation.
>
> | Grocery Store (582) |                | Patio (706)        |                 |
> |-|-|-|-|
> | **Biased**           | **Not Biased** | **Biased**          | **Not Biased**  |
> | motel                | fan            | restaurant          | head            |
> | library              | coffin         | school              | book            |
> | wedding              | factory        | house               | man             |
> | party                | sale           | pool                | the             |
> | restaurant           | factory        | residence           | republic        |
> | party                | distillery     |                     | tree            |
> | wedding              | market         |                     | window          |
> | motel                | supermarket    |                     | house           |
> | bakery               |                |                     | courtyard       |
> | supermarket          |                |                     | terrace         |
> |                      |                |                     | patio           |
>
>
>
>
> We will include these details in the revised Appendix and clarify how it complements the simpler last-token strategy used for Table 7.
>
>
> **[Suggestions: The results in Section 4.2 are a bit hard to parse… / There are a few typos that should be fixed…]**
>
> We will revise the results and captions in Section 4.2 and Tables 3–4 to improve clarity. Specifically, we will add the following explanation:
>
> “We generate 100 images per class and report the percentage of images that activate the target class.”
>
> We also thank the reviewer for spotting the typos in the loss term $\mathcal{L}_{\text{mask}}$ (page 9, lines 319 and 322), which will be corrected.

---

> > ### Comment · Reviewer_xK8N · 2025-08-02
> >
> > I thank the authors for their responses. They have addressed my concerns and clarified my confusions regarding the activation scores and the selection of top-k words. Integrating the explanations provided to all the reviewers should improve the paper's clarity and help readers better understand the proposed method.

---

### Official Review · Reviewer_jRsB · 2025-07-18

**Clarity:** 2
**Significance:** 3
**Originality:** 3
**Rating:** 5
**Confidence:** 4

**Summary:**

The paper introduces a novel, data-free framework, DEXTER, to generate global multimodal explanations for vision classifiers. It leverages diffusion models for image generation and large language models (LLMs) for textual reasoning, providing interpretability without requiring training data or labels. The paper evaluates DEXTER on multiple interpretability tasks including activation maximization, slice discovery, and bias explanation, using standard datasets like ImageNet, Waterbirds, CelebA, and FairFaces.

**Questions:**

1- During training, in each optimization step, how many inference steps were taken for LCM? Is it just 1 inference step?
2- How can Semantic CLIP-IQA achieves higher score than CLIP-IQA? Provide intuition. Does this mean some images are good photo of [CLASS] but is not a good photo?
3- Can you please provide comments on how to efficiently select feature neurons if they are not provided in a dataset?

**Ethical Concerns:**

["NO or VERY MINOR ethics concerns only"]

**Final Justification:**

The authors' responses have successfully addressed my confusion and concerns about the proposed framework.

Due to the acknowledged limitations of the proposed framework, in particular its generalization to other complex tasks or other visual classifiers, I couldn't increase my score.

**Limitations:**

### Possible additional limitations of the proposal
- **Dependence on foundation models:** Another possible limitation is that DEXTER requires VLM and Stable Diffusion need to have conceptual/semantics understanding of common objects. For example, when investigating chest-X-ray-image related classes but Stable Diffusion was not pretrained with Chest X-ray images. So
- **Generalization to other complex tasks or other visual classifiers:**
  + The current evaluation only experimented with one visual classifier, RobustResNet50, which fails to demonstrate the ability to generalize DEXTER performance on other visual classifiers.
  + The paper focuses on image classification models. It is unclear how well the approach would generalize to detection, segmentation, or multimodal tasks involving reasoning across modalities beyond vision and text.
  + The pipeline is customized for class-wise analysis, which may not scale well to instance-level explainability in domains like medical imaging.

**Quality:**

3

**Strengths And Weaknesses:**

## Strengths
### Contribution is novel
- DEXTER that extracts global, class-wise or feature neuron model explanations without any access to data or labels is highly novel. This advances the field of post-hoc interpretability significantly.
- DEXTER generates multimodal explanation combining text explanation, textual reasoning, and visual synthesis. This advances the explanations more human-interpretable than existing visual-only or textual-only approaches.
### Thorough Evaluation and Benchmarks
- DEXTER was evaluated on four well-motivated tasks using four corresponding datasets.
- Performance is validated with quantitative and qualitative metrics, user studies, and ablation studies.
### Correctness
- To the best of my knowledge, the proposal appears to be sound. Although DEXTER includes foundational models like BERT, StableDiffusion, and CLIP, but the whole process was designed such novel and well that the whole process is fully differentiable allowing to optimize soft prompt p which is the only learnable parameter.

## Weaknesses:
### Proposal's presentation can be improved
- The paper lacks of a formalization of generating bias reasoning and slice discovery. Bias reasoning and slide discovery are illustrated in Figure 1 as main contributions of the paper beside DEXTER. However, their current descriptions seem to be dataset-dependent as they are given only along with specific datasets in the Section Performance Analysis. Also, Slice Discovery was experimented only with class neuron? This will raise the question whether Slice Discovery can be applied for feature neuron or not?
- The selection of P in soft prompts was neither discussed nor analyzed. The advantages/disadvantages of P > 1 compared with P = 1 demand futher analyses.
- There is no discussion on how to determine key neural features efficiently. The paper only provided use cases where neural features are already included in the used dataset SalientImageNet.
- d(e) used in the formula n = f(d(e)) demonstrated that the diffusion model is applied one time per optimization step during training (i.e., the number of inference step for Stable Diffusion is 1). Is this correct? This demands further clarification.
- The design choice of loss function in Formula (2) needs discussion. Is that because if n_i is class neuron, 0 <= n_i < 1, the corresponding loss is at logarithm scale, and if n_i is feature neuron, n_i can be much greater, thus the corresponding loss is at linear scale?
- The paper did not discuss the potential issues of the randomness in each stable diffusion's inference step i.e., d(e) during training.
- o_i^{(C)} = Mo_i seems not correct. Should it be o_i^{(C)} = o_i M?
- The experimentation results showed that DEXTER prefers conceptual while DiffExplainer prefers perceptual. However, it is yet unclear for which of their design differences attributed for those observations.
- Formula (4) is not well-defined. The uses of subscriptions j and i in \Sum_{j=1}^N log s_{i, y_i} are confused.
- How can Semantic CLIP-IQA achieves higher score than CLIP-IQA? Provide intuition. Does this mean some images are good photo of [CLASS] but is not a good photo?

### Additional limitations of the proposal
- **Dependence on foundation models:** Another possible limitation is that DEXTER requires VLM and Stable Diffusion need to have conceptual/semantics understanding of common objects. For example, when investigating chest-X-ray-image related classes but Stable Diffusion was not pretrained with Chest X-ray images. So
- **Generalization to other complex tasks or other visual classifiers:**
  + The current evaluation only experimented with one visual classifier, RobustResNet50, which fails to demonstrate the ability to generalize DEXTER performance on other visual classifiers.
  + The paper focuses on image classification models. It is unclear how well the approach would generalize to detection, segmentation, or multimodal tasks involving reasoning across modalities beyond vision and text.
  + The pipeline is customized for class-wise analysis, which may not scale well to instance-level explainability in domains like medical imaging.

### Other minor comments:
- To the best of my knowledge, the target class should be one of the best pseudo-lables returned, and such pseudo-labels are not of interests. How to avoid this? Considering the example in Figure 1 where the investigation class is `bee`, how DEXTER avoids return the pseudo-labels as `bee` which is not of interests.
- The claim `DiffExplainer [31] introduced diffusion-based image generation guided by soft prompts, improving realism but at the cost of interpretability: soft prompts are continuous vectors with no semantic transparency. ` is not precise. Indeed, [31] includes section 4.4 `towards text explanations` to instruct how to provide textual explanations (eventhough this was not fully investigated in the paper)
- The textual explanations rely on LLM captions and summarization, which may introduce hallucinations—although mitigated, this remains a concern.

---

> ### Author Rebuttal · Authors · 2025-07-30
>
> We thank the reviewer for the careful evaluation of our work and the thoughtful comments. Below, we address the identified weaknesses (W) and limitations (L), providing clarifications and details to improve the manuscript’s clarity and rigor.
>
> **[W1 - Formalization of generating bias reasoning and slice discovery]**
>
> Appendices C and D already provide high-level explanations of the two methods. In the final version, we will include the following explicit formulations for bias reasoning and slice discovery, emphasizing that both are fully dataset-independent, relying only on model behavior without access to training data or labels.
>
> **Formalization of slice discovery**
>
> DEXTER performs slice discovery without using training data. Given a classifier f and a target class c, it optimizes a text prompt to extract a set of top-k class words
> $W_c = {w_{c1}, \ldots, w_{ck}}$.
> Each word $w_{ci}$ is encoded using CLIP’s text encoder to obtain an embedding $t_{ci}$, and the class prototype is defined as the average embedding:
> $\bar{t}c = \frac{1}{k} \sum t_{ci}$.
>
> At inference time, each image $x_j$ is encoded into $v_j$ using CLIP’s image encoder. The similarity between the image and the class is then computed as:
> $\text{CLIPScore}(x_j, c) = \text{similarity}(v_j, \bar{t}_c)$.
> The predicted class $\hat{y}_j$ is the one with the highest similarity.
>
> In binary classification settings, slices are defined by prediction outcomes:
> - If $\hat{y}_j = y_j$, the image is assigned to the unbiased slice.
> - Otherwise, it belongs to the biased slice.
>
> Prompt optimization does not require access to the dataset, which is only used for scoring. This approach aligns with prior definitions of slices (e.g., Kim et al.) as input subgroups where the classifier exhibits systematic bias. While DEXTER performs slice discovery using class neurons, it also supports feature neurons, allowing broader applications (see Sec. 4.2.1).
>
> **Formalization of bias reasoning**
>
> Given a classifier f and class c, we optimize a prompt $t^*$ such that a diffusion model generates images
> ${x_1, \ldots, x_M}$ that maximize the classifier’s output for c.
> Each generated image is captioned using a vision-language model g, producing descriptions $d_i = g(x_i)$.
> A language model h then summarizes the captions into a single report
> $r_c = h(d_1, \ldots, d_M)$, describing the features associated with class c and possible biases.
> This process is fully data-free: both image generation and reasoning depend only on the model’s behavior. As shown in Appendix D, the resulting explanations align well with those derived from real data.
>
> **[W2 - Number of P in soft prompts]**
>
> We set P = 1 to keep the setup minimal and stable. In DEXTER, the soft prompt guides BERT in selecting hard tokens via masked language modeling, rather than encoding semantics directly. Expressiveness comes from composing multi-token prompts (typically 5–6; see Appendix A), not from prompt dimensionality. Experiments with P > 1 show that increasing P expands the parameter space and weakens the classifier’s gradient signal, destabilizing optimization in our data-free setting, as shown by these results:
>
> | P   | Spurious       | Core          | Avg           |
> |-----|----------------|---------------|----------------|
> | 1   | **81.25 ± 18.99**  | **94.25 ± 4.38**  | **87.75 ± 11.68**  |
> | 3   | 75.00 ± 29.00  | 44.50 ± 35.89 | 59.75 ± 32.44  |
> | 5   | 73.50 ± 25.66  | 74.00 ± 42.75 | 73.75 ± 34.20  |
> | 10  | 21.25 ± 31.17  | 67.75 ± 40.01 | 44.50 ± 35.59  |
>
> **[W3 – Determining key neural features]**
>
> We used RobustResNet50 for neural feature selection due to the availability of pseudo-ground truth from Salient ImageNet. For other classifiers (Appendix D.3), we ranked penultimate-layer neurons by their weights to each class and selected the top 5, mirroring the Salient ImageNet method but without any training data. This highlights DEXTER’s fully data-independent design.
>
> **[W4 -  $n = f(d(e))$ --> diffusion model is applied once]**
>
> While the diffusion model is invoked once per optimization step, its diffusion process consists of 4 denoising steps. We will include this detail in Appendix A.
>
> **[W5 - Formula (2)]**
>
> Correct. For class neurons (bounded in $[0,1]$), we use $\log(n_i)$ to encourage confident predictions. For unbounded feature neurons, we use a linear objective ($–n_i$) to directly maximize activation.
>
> **[W6 - Randomness in diffusion's inference step]**
>
> This is an important point. To assess the stability of Stable Diffusion, we performed intra-class evaluation using DEXTER’s final prompt. Across three independent runs (100 images each), we measured activation scores with the target model. If generation were unstable, scores would vary significantly; instead, consistent results indicate stability. We report mean and standard deviation for 8 randomly selected classes (4 core, 4 spurious) across three independent runs:
>
> | Type      | Class | Avg              |
> |-----------|-------|------------------|
> | Spurious  | 890   | 55.33 ± 2.49     |
> |           | 795   | 79.00 ± 1.63     |
> |           | 655   | 96.66 ± 0.94     |
> |           | 706   | 100.0 ± 0.00     |
> |           |       |                  |
> | Core      | 291   | 100.0 ± 0.00     |
> |           | 486   | 95.66 ± 1.24     |
> |           | 514   | 86.66 ± 1.24     |
> |           | 624   | 95.66 ± 1.88     |
>
>
> **[W7: $o_i^{(C)} = Mo_i$ incorrect]**
>
> Right. The formula is $o_i^{(C)} = o_i M$ and will be corrected in the final revision.
>
> **[W8 - Explanation for DEXTER preferring conceptual while DiffExplainer perceptual ones]**
>
> DEXTER conditions generation via text: it optimizes a BERT-based prompt decoded into a multi-token hard prompt for a text-to-image diffusion model, yielding semantically rich outputs. DiffExplainer instead optimizes a soft prompt in the diffusion latent space, without language supervision, often producing low-level patterns (e.g., texture, color) with less semantic coherence. This difference explains why DEXTER generates more conceptually grounded images, as shown in Figure 3 and confirmed by our user study (Appendices B.2, F).
>
> **[W9 - Formula (4)]**
>
> We have corrected the indices in the formula:
> $$
> \mathcal{L} = \sum_{k=1}^K l_{\text{act}}(n_k) - \sum_{i=1}^N  \log s_{i,y_i}
> $$
>
> **[W10 - Semantic CLIP-IQA vs CLIP-IQA scores]**
>
> The higher Semantic CLIP-IQA scores are not due to better image quality, but due to the use of class-specific prompts (e.g., “a good photo of a [CLASS]”), which enhance semantic alignment. In contrast, standard CLIP-IQA uses a generic prompt, lacking class-level grounding. As a result, the two metrics are not directly comparable, as prompt design influences the similarity evaluation.
>
> **[L1 - Dependence on foundation models]**
>
> While explanation quality depends on the underlying foundation models, DEXTER’s modular design supports easy adaptation to other domains. In medical imaging, for example, it can integrate models like PubMedBERT for prompts and diffusion models trained on X-rays or MRIs. This requires careful model selection to align medical terms with visual content, but does not alter DEXTER’s core architecture or functioning.
>
> **[L2 - DEXTER Generalization]**
>
> The reviewer raised three important issues: (1) generalization to other visual classifiers, (2) applicability to tasks beyond classification, and (3) scalability to instance-level explainability.
>
> 1) We chose RobustResNet50 for main experiments due to biased annotations in Salient ImageNet. However, as shown in Appendix D.3, we also applied DEXTER to AlexNet, ResNet50, and ViT. Results show consistent ability to investigate model-specific bias across both CNNs and transformers. For instance, Jeep shows bias in AlexNet/ResNet50 but not in ViT/RobustResNet, while Snorkel shows bias across all models.
>
> 2) DEXTER was designed for image classification, producing class-level global explanations. We acknowledge that tasks like detection or segmentation involve instance-level, spatial outputs. While not yet applied to these tasks, DEXTER’s modular design could support extensions—e.g., by conditioning on predicted regions or masks. Enabling such reasoning would require additional mechanisms and is a promising direction for future work.
>
> 3) DEXTER was designed to provide class-wise, global interpretable insights into the decision-making process of a visual classifier. By construction, it does not target per-sample (instance-level) explainability and may overlook fine-grained variations that are better captured by local methods (e.g., GradCAM, IG, LIME-X). However, while it doesn’t produce per-sample explanations, combining it with slice discovery enables subgroup-level conceptual explanations, thus offering a path toward multi-level interpretability.
>
> **[Minor comments]**
>
> Due to space constraints, we address minor comments 1–3 collectively.
>
> 1) The words generated by DEXTER indicate whether a class relies on spurious features. Core classes yield words closely aligned with the label, while spurious ones produce unrelated or contextual terms (e.g., “snow” or “trees” for dogsled). To capture this, we use multiple mask tokens, enabling richer, more informative prompts. Core classes show token convergence; spurious ones diverge. Recovering the class label suggests the classifier focuses on meaningful, non-spurious features.
>
> 2) We agree and will revise the statement. While DiffExplainer [31] briefly mentions text explanations (Sec. 4.4), DEXTER extends this with a full pipeline that optimizes prompts for both image generation and semantic explanation. We will clarify this connection in the final version.
>
> 3) We acknowledge the potential for LLM hallucinations. To address this, we implemented mitigation strategies and evaluated consistency in Appendix G.2. By linking visual activation cues to generated text and measuring alignment (Table 10), we show that DEXTER’s explanations are generally robust and reliable to LLM hallucinations.

---

> > ### Comment · Area_Chair_ZPUc · 2025-08-05
> >
> > Dear Reviewer jRsB,
> >
> > this is a reminder to please read the author's rebuttal and respond if there are points still left to resolve. Even if not, please provide a short commend and then provide your mandatory acknowledgement.
> >
> > Thank you very much!
> >
> > Best wishes
> >
> > AC

---

> ### Comment · Reviewer_jRsB · 2025-08-06
> **Thanks for the authors' responses**
>
> Thank the authors for your detailed responses. They have addressed successfully most of my concerns. I hope the clarification and the additional important contents in the responses can be well integrated into revised version of the paper. Due to the limitations of the proposed framework included in my reviews, I couldn't increase my score.
>
> Thanks

---

### Comment · Area_Chair_ZPUc · 2025-08-02
**Reviewer-Author-Discussions**

Dear all,

I wish to thank all the reviewers for their work and the authors for their detailed response. I would encourage the reviewers to consider the rebuttals and check whether the points are addressed.

Best wishes

AC

---

### Decision · Program_Chairs · 2025-09-17

**Decision:**

Accept (spotlight)

**Comment:**

I wish to thank all reviewers and the authors for their work and the extensive discussion.

**Strengths:**

- The core idea of generating global explanations of an image classifier without access to the training data (just the name of the class to be explained) is striking in its simplicity and wide-ranging applicability. Multiple reviewers highlighted the core contribution and I would add that this contribution is likely of wide interest in the NeurIPS community.
- Multiple reviewers also highlighted the flexibility of the approch in terms of easily exchangeable modules (Reviewer xK8N) or adjustability to different tasks (Reviewer cbH4).
- All reviewers emphasized the quality of the evaluation with abundant data sets, tasks, and baselines, as well as a user study.

**Weaknesses:**

- Multiple methodological issues were raised by the reviewers (with Reviewr cbH4 in particular calling the method illustration 'unorganized' and mentioning 'insufficient details for some key steps'). Fortunately, thanks to extensive replies by the authors during the discussion phase (including new results and tables) these concerns could be addressed during the discussion phase.
- One particular concern by Reviewer jRsB was the high computational demand of generating explanations. While the authors agreed that the prompt optimization is rather demanding with ca. 10 min. per class, each single explanation generation is efficient (ca. 15 seconds). This satisfied the reviewers but should probably clarified as 'price tag' of the method.

Overall, after discussions, all reviewers consistently voted to 'accept' and I concur.

I would also concur with Reviewer cbH4 that the clarifications during discussion should be integrated into the main paper for the camera ready.